

# Updated MISR Dark-Water Research Aerosol Retrieval Algorithm Part 1: Empirical Calibration Corrections and Coupled 1.1 km Ocean-Surface Chlorophyll-a Retrievals

James A. Limbacher[1,2] and Ralph A. Kahn[1]

[1]Earth Science Division, NASA Goddard Space Flight Center, Greenbelt, 20771, USA
[2]Science Systems and Applications Inc., Lanham, 20706, USA

*Correspondence to*: Ralph A. Kahn (ralph.a.kahn@nasa.gov)

**Abstract.** As aerosol amount and type are key factors in the "atmospheric correction" required for remote-sensing
chlorophyll-a concentration (*Chl*) retrievals, the Multi-Angle Imaging SpectroRadiometer (MISR) can contribute to ocean
color analysis despite a lack of spectral channels optimized for this application. Conversely, an improved ocean-surface
constraint should also improve MISR aerosol-type products, especially spectral single-scattering albedo retrievals. We
develop and apply new calibration corrections to the MISR top-of-atmosphere (TOA) reflectance data, and introduce a self-
consistent retrieval of *Chl* together with aerosol over dark water. The calibration corrections include: a modified stray-light
model based on comparison with coincident MODerate-resolution Imaging Spectroradiometer (MODIS) Terra data, and
trend analysis using MISR TOA bidirectional reflectance factors (BRFs) over three pseudo-invariant desert sites. The trend
analysis shows that MISR radiometric sensitivity decreased by up to 2 percent for MISR spectral bands between January
2002 and December 2014.

After applying calibration corrections, we run the MISR Research Retrieval Algorithm (RA) to validate the MISR RA-
retrieved *Chl*, and analyze both the MISR and corresponding MODIS-Terra values compared to a set of 49 collocated
SeaBASS in situ observations, constrained to $Chl_{in\,situ} < 1.5$ mgm$^{-3}$. Statistically, compared to the validation data, MODIS
demonstrates a higher correlation coefficient (r) of 0.91 vs. 0.86 for MISR, a lower root-mean-squared-error (RMSE) of 0.25
vs. 0.22, but a higher median absolute error (MAE) of 0.14 vs. 0.10. Because 49 data points are insufficient to draw strong
conclusions, we also compare MODIS-Terra and MISR RA *Chl* statistically, over broader regions. With about 1.5 million
MISR-MODIS collocations having MODIS *Chl* < 1.5: r=0.96, MAE=0.09, and RMSE=0.15. MISR-MODIS agreement is
substantially better than the 49-data-point MODIS-SeaBASS comparison, indicating that MISR *Chl* retrievals might
complement MODIS, especially after further upgrades are made to the MISR RA ocean color model.

The new dark water aerosol/*Chl* RA can retrieve *Chl* in low-*Chl* (<1.5), case I waters, independent of other imagers such as
MODIS, via a largely physical algorithm, compared to the commonly applied statistical ones. At a minimum, MISR's
unique multi-angular data can better constrain aerosol type, helping reduce uncertainties in the MODIS Terra Ocean color





retrieval, and suggesting how a joint MISR-MODIS over-ocean algorithm might exceed the capabilities of either instrument alone.

## 1 Introduction

Among the geophysical quantities routinely produced from the NASA Earth Observing System's Multi-angle Imaging
SpectroRadiometer (MISR) instrument are aerosol optical depth (AOD) and aerosol type. MISR's unique multi-angle, multi-spectral radiance data sample air-mass-factors ranging systematically from one to three, making AOD retrieval possible even over bright desert surfaces, and improving retrieval sensitivity at low AOD compared to single-view instruments. In low-AOD situations, which are common over ocean, poor representation of the surface reflectance can limit aerosol retrieval accuracy, as the relative contribution of ocean under-light can be large, especially at shorter wavelengths.
Therefore, applying a physical retrieval to constrain the ocean surface reflectance, of interest in itself as an indicator of ocean biological activity and its impact on the global carbon cycle (e.g., Behrenfeld et al., 2006), should also reduce the uncertainties in the concomitant aerosol retrievals.

A second factor directly affecting the quality of almost every MISR geophysical data product is the accuracy of the instrument's radiometric calibration. Calibration includes determination of (1) the absolute radiometric scale, as well as (2)
the relative band-to-band response among the four MISR spectral bands, (3) camera-to-camera response among the nine MISR cameras, (4) flat-fielding across the MISR imagery, and (5) temporal trends in these quantities. Considerable effort has been expended to assess MISR radiometric calibration and to meet the standards of approximately 3% absolute and 1% channel-to-channel, established pre-launch. This work involved pre-launch laboratory studies (Bruegge et al., 1999), on-board-calibrator analysis and lunar calibration, along with vicarious calibration over bright land targets (Bruegge et al., 2004;
2007; 2014), symmetry tests comparing the forward and aft-viewing cameras across the solar equator (Diner et al., 2004), and over-ocean dark target vicarious calibration (Kahn et al., 2005). Cross-calibration analysis has been performed over bright and dark land and ocean surfaces with the MODerate-resolution Imaging Spectroradiometer (MODIS), that flies aboard the Terra satellite with MISR (Lyapustin et al., 2007), and MODIS combined with the MEdium Resolution Imaging Spectrometer (MERIS), the airborne AirMISR instrument, and the LandSat-7 ETM+ (Bruegge et al., 2007), and the
Polarization and Directionality of the Earth's Reflectances-2 (POLDER-2) (Lallart et al., 2008). A synthesis of much of this work is given in Bruegge et al. (2014).

As the MISR data record now exceeds 16 years of near-global coverage about once per week, the advantages of further refining the MISR calibration have increased multifold. This applies to determining AOD trends, and is especially true in the context of MISR's unique ability to retrieve aerosol type (Kahn and Gaitley, 2015). In addition to AOD and aerosol
type, retrievals of ocean bio-optical properties from space are extremely sensitive to the calibration of the instrument, because only 5 to 20% of the top-of-atmosphere (TOA) reflected signal in the blue and green spectral bands, where ocean color is retrieved, arises from scattering related to ocean under-light (e.g., Gordon and Wang, 1994). To retrieve this signal,





the following must be properly accounted for: (1) molecular (Rayleigh) scattering, (2) molecular absorption, (3) scattering from atmospheric aerosols, (4) absorption from atmospheric aerosols, (5) reflection from foam and whitecaps, and (6) Fresnel reflection (glint) from the ocean surface. Of course, the aerosol and molecular signals include both an atmospheric path radiance term and a term with at least 1 surface reflection. In earlier work, we used image analysis, including

comparisons with coincident MODIS observations, to identify empirical relationships that correct anomalies exhibiting spatial structure in high-contrast scenes, an aggregate of "ghosting" light reflections (or stray-light) within the cameras. We make minor adjustments to our earlier ghosting and flat-fielding (CCD detector-based gain errors, which will show up as across-track biases in reflectance) corrections for the results presented in this paper, but the underlying work for these corrections is found in Limbacher and Kahn (2015). In the course of this analysis, we also observed some systematic,

temporal drift in the measured reflectances. Addressing this calibration trend is a major focus of the current paper, as along with highlighting our ability to retrieve chlorophyll-a (*Chl*) and aerosol amount/type self-consistently with the corrected MISR reflectances, using an improved version of the MISR dark water MISR research algorithm (RA). The paper is organized as follows: section 2 reviews the datasets used in our analysis and the methodology adopted, section 3 addresses the observed temporal trends in MISR radiometric calibration from 2002 to 2015, section 4 presents the *Chl* retrievals and

initial validation of the results, and conclusions are given in section 5.

## 2 Comparison datasets and methodology

### 2.1 The MISR Research Algorithm, with Enhanced Ocean Reflectance Model

An in-depth description of the current RA can be found in Limbacher and Kahn (2014; 2015). Briefly, the algorithm finds the set of aerosol optical models, and associated aerosol amounts, that minimize the difference between the observed TOA

reflectances (identical to BRFs described in section 3, but without the solar-zenith angle normalization) and simulated values that are stored in a look-up table (LUT). The simulated ocean surface is modeled as a black, isotropic (wind-speed-dependent only) Fresnel-reflector, with whitecap reflectance included. Additionally, adjustments to the whitecap reflectance, and under-light due to molecular scattering, colored dissolved organic matter (CDOM) and *Chl* were previously modeled using wind and ocean-color constraints from the Cross-Calibrated Multi-Platform (CCMP; Atlas et al., 2011) and GlobColour

(Barrot et al., 2010) products, respectively, and from climatology where these products were unavailable.

For the current analysis, we continue using CCMP data for 10-meter wind speed (where available, otherwise we use the MISR Standard Algorithm (SA) wind data), and we set the surface pressure to 1013.25 mb, as we find a number of cases where the MISR Standard Product surface pressure over ocean is aliased from nearby mountains. Additionally, we now use all four spectral bands to simultaneously retrieve aerosol and *Chl*, with equal weighting, whereas the SA (Martonchik et al.,

2009) and past versions of the RA used only the red and NIR bands (except at high AOD), where the ocean surface is darkest. However, empirical weighting is applied to mitigate the effects of sun glint, and different uncertainties are assigned to the 36 MISR channels when evaluating the $\chi^2$ acceptance criteria, as discussed below. Generally, the near-nadir views and





shorter wavelength bands contain more information about the surface, whereas the steeper views tend to provide greater constraint on the atmospheric aerosols. A refinement to the algorithm, not implemented here, would be to separate the *Chl* and aerosol retrievals, so we can weight contributions from each channel in a manner that reflects the differences in information content. CDOM absorption is assumed to co-vary with *Chl* (Morel and Gentili, 2009). Relationships connecting

*Chl* to absorption and back-scattering coefficients can be found in many places; the ones we used (Chen et al., 2010; Devred et al., 2006; Morel and Prieur, 1977; Morel 1988) are summarized in Sayer et al., (2010). For our ocean under-light model, we modify the absorption of light by seawater for the blue spectral band from the Morel and Prieur (1977), which was used previously in the RA, to more recent results from Lee et al., (2015).

The following equation gives a bidirectional water-leaving radiance:

$$L_w^+\left(\mu_0,\mu,\Delta\phi,\lambda,WS,\tau,mix,Chl\right) = E_d\left(\mu_0,\lambda,\tau,mix\right) * \Re\left(\mu,WS\right) * \left(\frac{b_b\left(\lambda,Chl\right)}{a\left(\lambda,Chl\right)}\right) * \frac{f}{Q}\left(\mu_0,\mu,\Delta\phi,\lambda,Chl\right) \qquad (1)$$

The following explanation of the terms in equation (1) is basically a summary of Morel et al. (2002), which is also where the LUT for f/Q and $\Re$ was obtained. Variable dependences are included here only if they are given in the Morel et al. (2002) LUT.

- $L_w^+$ represents the water-leaving radiance, which is the upward-directed radiance just above the water surface (excluding sun-glint). It is a function of the cosine of the solar zenith angle ($\mu_0$), the cosine of the view (camera) zenith angle ($\mu$), the relative azimuth between the sun and the sensor ($\Delta\phi$), the wavelength ($\lambda$), the wind speed (*WS*), the total column optical depth ($\tau$), aerosol optical model (*mix*), and Chlorophyll-a concentration (*Chl*).
- $E_d$ represents the downward directed solar irradiance at the bottom of the atmosphere.
- $\Re$ is a reflectance factor, the product of two effects: the fraction of the downward directed bottom-of-atmosphere irradiance ($E_d$) transmitted through the air-sea interface, and the fraction of the upward directed radiance from just beneath the air sea interface transmitted through the interface.
- $b_b$ represents the total backscattering coefficient of the water plus other material within the water.
- *a* represents the total absorption coefficient of the water plus the other material within the water.
- *f* represents an empirical correction to the ratio of the backscattering to absorption (essentially a modification to the upward directed under-light irradiance).
- *Q* represents the ratio of the upward-directed irradiance to radiance just below the air-sea interface. This term (along with *f*) is responsible for creating the directional dependence of the under-light on solar and viewing
geometry.



$L_w^+$ is multiplied by the transmittance from the bottom of atmosphere to the camera ($T_{a,up)}$) to get the surface contribution to the TOA reflectances. Because the integrated water-leaving radiance, $L_w^+ \ll E_d$ (i.e., the under-light albedo is small), the probability that a photon will be multiply reflected due to under-light is very small, regardless of atmospheric loading, and is ignored, given other, larger uncertainties in the algorithm. However, multiple surface reflections due to sun-glint and
whitecaps are directly accounted for in the radiative transfer code.

Initial processing of the MISR radiances includes adjusting and applying our ghosting parameterization, and correcting for flat-fielding and for temporal degradation in the calibration (see Section 3 below). We then revise the band-to-band calibration by increasing the red reflectance 0.75% and decreasing the near-infrared (NIR) reflectance 0.75%, adjustments
that are within the calibration uncertainty and are required to match a global set of coincident, spectral aerosol optical depth validation data (*Limbacher and Kahn,* 2014; 2015). We also apply corrections to the radiance data to smooth out apparent anomalies in the instrument gain, based on Bruegge et al. (in preparation).

As we aim to extract both surface and aerosol information from the MISR data, we apply new camera weights when
calculating the $\chi^2$ test variables used to assess the agreement between the observed reflectances and those derived for various aerosol component and mixture options. In the SA and previous versions of the RA, a glitter mask was applied arbitrarily to all cameras viewing within 40˚ of the specular direction. Instead, we now use a combination of glitter-angle and Rayleigh NIR reflectance, calculated assuming $Chl = 0.1$ mg/m$^3$ to assess glint-contamination in each camera. The new camera weights are the product of the following two empirically derived equations:

$$\rho_{weight,i} = 1.0 - min\left\{max\left[\frac{(\mu_i * \rho_{model,i}^{NIR} - 0.0075)}{0.0125 - 0.0075}, 0.0\right], 1.0\right\}, \qquad (2)$$

$$glitter_{weight,i} = min\left\{max\left[\frac{(G_i - 25.0)}{40.0 - 25.0}, 0.0\right], 1.0\right\} \qquad (3)$$

Here $\rho_{model}^{NIR}$ represents the modeled NIR Rayleigh reflectance over an ocean surface for a particular MISR camera (*i*), and *G*
is the glitter angle relative to the same MISR camera. Equation 2 returns a value of unity if the modeled NIR reflectance is ≤0.0075, decreasing linearly to zero if the modeled reflectance is ≥0.0125. Similarly, Equation 3 produces a value of unity if the glitter angle ≥40˚, decreasing linearly to zero for G ≤ 25˚. The product of these weights provides better glint masking than using an arbitrary cutoff, and the quality of these new weights should improve with the quality of the input wind speed data.


The aerosol/*Chl* retrieval process is summarized as a flow chart in Figure 1. Essentially, the *Chl* retrieval can be thought of as an inversion of our ocean under-light model (i.e., instead of prescribing *Chl*, we retrieve it). The overall aim is to derive AOD and *Chl* over 1.1 km retrieval regions, conditioned on aerosol-type mixtures that produce TOA radiances that meet





certain $\chi^2$ criteria. In the current study, we compare the MISR RA *Chl* retrievals, after all MISR calibration corrections are applied, to coincident validation data taken at the surface, and also identify the impact the refined ocean surface model has on the retrieved aerosol results. In the RA pre-processing, all MISR L1B2 reflectance data are first averaged to 1.1 km. The reflectances are then rotated to the L1B1 format as described in Limbacher and Kahn (2015), and updated stray-light and

flat-fielding corrections are applied before being rotated back to L1B2 format. Compared to Limbacher and Kahn (2015), we modify the stray-light corrections in the following way:

- The primary ghost term has been divided into a discrete ghosting component (reflected images of features in the scene) and an unstructured veiling-light component.

  - This revised primary ghost has a band-and-camera-dependent along-track offset applied, as indicated by
10          MISR lunar observations acquired on 14 April 2003 (e.g., Bruegge et al., 2004).

  - The primary ghost image is also stretched/squeezed across-track (for the near-nadir "A" cameras only), based on further comparisons with MODIS Terra, following the same approach as our earlier work.

- Via ray tracing, it was found that the "secondary ghosting" term in Limbacher and Kahn (2015) distributes light uniformly from the left- or right-most ~1/3 of the scene to the remainder of that half of the scene (Ab Davis,
personal communication, 2016), and the correction has been modified accordingly.

- All stray-light terms are now represented as convolutions, which are much quicker to compute than applying the functions pixel-by-pixel as was done in our earlier work.

- The magnitudes of all stray-light terms have been adjusted as a result of adding the unstructured veiling-light component.

- The stray-light model for the AN camera (all four bands) is used for all off-nadir cameras. Only the along-track offset and primary ghost stretching are varied by camera.

Figure 2 illustrates the impact of including under-light in the MISR RA, for the blue and green-band retrieval TOA reflectance results. For this figure, MISR aerosol retrievals over dark water were performed using the angular data only for the NIR band, because the ocean surface tends to be darkest at this wavelength, as under-light makes its smallest spectral

contribution. When the retrieved aerosol properties are used in the forward radiative transfer model to simulate the MISR top-of-atmosphere (TOA) reflectances in the blue and green bands, but under-light is not included, there are large discrepancies in the modeled TOA reflectances compared to the original MISR observations (Figure 2, top two panels). However, when under-light is accounted for in the simulations (in this illustration using coincident MODIS Terra *Chl* as input), the biases are substantially reduced, as shown in the lower two panels of Figure 2. As the MODIS-constrained *Chl*

was included when the aerosol retrieval was performed (with the multi-angle NIR data only), this example demonstrates the magnitude of the surface contribution to the TOA reflectances in the blue and green spectral bands. If surface contributions are not explicitly included, the aerosol retrievals would be skewed, and the spectral dependence of the anomaly would have a large effect on the derived aerosol type (e.g., Kahn and Gaitley, 2015), especially when the blue or green bands are included in the aerosol retrieval. In Section 4 and supplemental material we demonstrate the use of MISR to constrain *Chl* self-





consistently with the retrieval of aerosol over ocean. However, we first refine the calibration of MISR, as described in section 3.

**2.1 MODIS Terra top-of-atmosphere reflectances**

As in Limbacher and Kahn (2014), MODIS-Terra equivalent reflectance data are used as a baseline to compare against

MISR, especially for the near-nadir cameras. We use the latest MODIS collection 6 TOA reflectances (Sun et al., 2012) with additional corrections implemented via an algorithm provided by Alexi Lyapustin (Lyapustin et al., 2014; elaborated in Limbacher and Kahn, 2015). Primarily, we are interested in the following MODIS bands: 9 (443 nm, as compared to MISR's 446 nm blue), 4 (555 nm, as compared to MISR's 558 nm green), an average of bands 13 and 14 (effectively 672 nm, as compared to MISR's 672 nm red), and 2 (856 nm, as compared to MISR's 866 nm NIR). In the current study, MODIS

reflectances are used only to remove flat-fielding artifacts in the MISR imagery and to make modifications to the ghosting parameterization described in Limbacher and Kahn (2015), so the absolute calibration accuracy of MODIS is not critical here. For the flat-fielding characterization, we select only low-contrast scenes, where ghosting artifacts are minimal, and we then normalize the mean MISR-MODIS ratios for the entire scene to unity. For the ghosting modifications, we normalize the MISR-MODIS ratios to an area of little contrast, where stray light is unlikely to be a problem. The most critical

assumptions are that MODIS swath-edge and scan-angle issues are minimal for the scenes of interest, and that pixel-to-pixel relative precision is high. Fortunately, because the MISR swath samples about 380 km around the center of the 2,300 km MODIS swath, the effects of MODIS swath-edge and scan-angle artifacts on the coincident data are minimal.

**2.3 The SeaWiFS Bio-optical Archive and Storage System (SeaBASS) data set**

The SeaBASS dataset (Werdell et al., 2002) was originally developed to compare products retrieved from sensors such as the

Sea-viewing Wide Field-of-view Sensor (SeaWiFS) and MODIS with in situ bio-optical observations. We use SeaBASS chlorophyll validation data generated either by fluorometry or by high-performance liquid chromatography (HPLC). Uncertainties for HPLC and fluorometry *Chl* measurements are 5% and 8%, respectively (Heukelem et al., 2002). If HPLC (*Chl*) and fluorometry (*Chl*) data were acquired at the same location and time, we use the HPLC (*Chl*) data; otherwise we use whichever data are available. Because the MISR Standard Algorithm does not retrieve *Chl*, the MISR-SeaBASS

coincidences were found by locating MODIS-Terra validation matchups (Bailey and Werdell, 2006) and setting the viewing-zenith angle maximum to 16°, which corresponds to the edge of the MISR nadir (AN) camera field-of-view. In addition, (1) minimum sea floor depth was set to 30 meters to mitigate errors due to sea floor reflections, especially in the blue band, (2) maximum wind speed was set to 7 m/s to avoid whitecaps (eliminating ~25% of data), (3) maximum solar zenith angle was set to 70°, (4) maximum coefficient of variation for MODIS *Chl* was set to 0.15, (5) maximum SeaBASS-MISR time

difference was set to 3 hours, and (6) minimum number of valid MODIS pixels was set to 25%, resulting in 75 coincidences that have valid MISR aerosol/*Chl* retrievals. Of these 75 coincidences, only about 50 correspond to *Chl* < 1.5 and also have at least one valid MISR RA retrieval in a 5.5 x 5.5 km area surrounding the SeaBASS station passing our quality tests.



### 2.4 MODIS Terra Chlorophyll-a

Although we validate our *Chl* retrieval against the SeaBASS dataset for *Chl* <1.5, we also cross-compare our *Chl* results with MODIS-Terra (OBPG, OB.DAAC; 2014) to increase the number of coincidences (especially needed for *Chl* < 1.5), and because MISR and MODIS share a common platform. This ensures that the solar geometry is the same for MODIS and

MISR, and minimizes potential collocation errors. To do this, we compare MISR RA-retrieved *Chl* with the corresponding MODIS Terra retrieved values (Hu et al., 2012). Details of the algorithm used to generate the MODIS data can be found at http://oceancolor.gsfc.nasa.gov/cms/atbd/chlor_a. Briefly, a training dataset containing collocated in situ *Chl* and spectral water leaving radiance ($L_w^+$) is used to empirically relate the ratio of blue-to-green MODIS $L_w^+$ to near-surface *Chl* (Werdell and Bailey, 2005). This same relationship is then used to retrieve MODIS *Chl* elsewhere, although the quality of the result

also depends in part on the quality of the associated atmospheric correction (e.g., Kahn et al., 2016).

### 2.5 The AErosol Robotic Network (AERONET)

Although the main purpose of this paper is to demonstrate and validate our *Chl* retrieval, we also compare the new algorithm against AErosol RObotic NETwork (AERONET) observations (in the supplemental material) for a few selected scenes. AERONET sun photometers (Holben et al., 1998) provide very accurate measurements of AOD (Eck et al., 1999) and

Ångström exponent (ANG). The almucantar inversions (Dubovik and King, 2000) can provide particle sphericity (Dubovik et al., 2006; which we convert to fraction mid-visible AOD assigned to non-spherical particles, or Fr. Non-sph), and aerosol single scattering albedos (SSAs), provided the aerosol loading is high, the scattering angle range for the inversions is large, and the aerosol is relatively uniform over the range of view angles used for the inversion (Holben et al., AERONET's Version 2.0 quality assurance criteria).

### 3 Temporal Trends in the MISR Calibration

As all aspects of the MISR calibration, in addition to correction for high-contrast-scene artifacts, can affect retrieval products such as aerosol type and ocean surface properties (Limbacher and Kahn, 2015), we identify here temporal trends in the instrument calibration, again using an empirical image-analysis approach. Bruegge et al. (2014) identified temporal trends in

the MISR bidirectional reflectance factor (BRF, computed as described in Step 1a. below) data, based on a time-series of mean BRFs for a region approximately 10 × 20 high-resolution (275 m) MISR pixels (roughly 2.5 × 6 km) in size, centered at (27.21˚ N; 26.10˚ E) within the Egypt-1 stable desert site. Although this site is stable over time, we adopt a different methodology, similar to Lyapustin et al. (2014), using average BRFs over larger areas at three stable desert sites (Egypt-1, Libya-1, and Libya-4). Both techniques are valid, but given the homogeneity of the selected sites, we limit geo-location

error by averaging, and we reduce the influence of clouds by selecting the median BRF pixel from each case.





The first challenge to performing the temporal-trend analysis is finding suitable homogeneous regions. The following was done to select study regions within each of the three sites: (a) The spectral coefficients of variation (standard deviation divided by the mean) were calculated for rolling 50 × 50 pixel patches, in each spectral band of the nadir camera, for three or more orbits. (b) The 50 × 50 pixel patch having the smallest maximum coefficient of variation among the selected orbits and

the four spectral band values was chosen for subsequent time-series analysis. The central coordinates for the sites selected are: Egypt-1 (26.62° N, 26.18° E), Libya-1 (24.73° N, 13.52° E), and Libya-4 (28.77° N, 23.50° E). Information about these calibration sites can be found at http://calval.cr.usgs.gov/rst-resources/sites_catalog/radiometric-sites/test-site-gallery/. The central coordinate of each study site is imaged repeatedly by MISR along at least two distinct paths having different sub-spacecraft ground tracks, and therefore different viewing geometries at the site. (A "path" is one of 233 ground tracks that

the Terra satellite covers, repeatedly, every 16 days.) So the following procedure was applied separately to each path and camera (6 paths × 9 cameras), for data acquired between January 1, 2002, and December 31, 2014, giving 13 full years of MISR data. (Prior to January 1 2002, the spacecraft equator-crossing time was not yet stable, so viewing geometry varied too much for this time-series analysis.) All observations of each site, about four per month, were initially included. Note that we also apply flat-fielding corrections (Limbacher and Kahn, 2015), and additional corrections to the radiance data to

smooth the instrument gain temporal samples (Carol Bruegge, personal communication, 2016).

1) **Calculate median patch reflectance for each orbit**

    a. Perform Earth-Sun and solar zenith normalization according to: $BRF = L*([\pi \times D^2] / [E(i) \times \cos(SZA)])$, where L is the top-of-atmosphere radiance, D is the sun-Earth distance in AU, E(i) is the band-weighted exo-atmospheric solar irradiance for band (i), and SZA is the solar zenith angle.

b. Calculate the median (and mean) BRF and standard deviation over a region 25 km in radius surrounding the central latitude/longitude coordinate.

    c. If the wavelength-maximized coefficient of variation is less than 0.02, save the median BRF for use in the time series, otherwise discard the data.

Median BRF values for at least 193 orbits, and up to 229 orbits, were retained for all 6 paths, 4 spectral bands, and 9

cameras at this step.

2) **Remove outliers for each path/site and spectral band**

    a. Arrange the saved median BRFs by acquisition date, fit a line to the values, and subtract the linear trend from the data.

    b. Aggregate the data by day-of-year (DOY) and smooth the sorted, de-trended BRFs using a 21-point (i.e.,

±10 data point) rolling average. (The data are sufficiently dense that replacing each data point with the mean of 21 points does not create significant artifacts in the time-series.)

    c. Identify BRFs that fall outside 2σ from time series.

    d. Remove the identified outliers from the original data.





This step removed 3-14% of data outliers from each time-series.

3) **De-seasonalize the data for each site and spectral band**

    a. Fit a line to the original, time-ordered BRFs, with outliers removed, and linearly de-trend the data.

    b. Re-aggregate the data by DOY and smooth the BRFs again using a 21-point (±10 data point) rolling average.

    c. Rearrange the data by time and add back the linear trend from Step 3a.

Step 3 is illustrated in Figure 3 for the Libya-4 site.

4) **Normalize the data**

    a. Normalize the data so the time-series mean for each spectral band at each site is 1.0, allowing data from multiple sites and paths to be compared.

The result is 216 normalized time-series, one for each MISR camera and band, for each of two paths at three sites.

    b. These time-series are then aggregated across all paths to produce 36 time series, one for each MISR channel (Figure 4).

The linear percent change per decade and its 95[th] percent confidence interval are then calculated for each channel, and the results are presented in Table 1 and Figure 5. The trends are all negative, as might be expected due to sensor degradation over time. They are smallest in the blue band for all but the forward-viewing 70.5˚ (Df) and 60.0˚ (Cf) cameras, smallest for the aft-viewing 70.5˚ (Da) and 60.0˚ (Ca) cameras for all bands except the NIR, and largest for the An, and 26.1˚ forward (Af) and aft-viewing (Aa) cameras. The largest drift overall is about -1.5% per decade for the An camera red and NIR bands, and the uncertainty in these results ranges from ~0.1 % per decade to ~0.4 % per decade, depending on wavelength and camera. The apparent stability of the MISR blue band is probably due to the use of the blue diode to assess degradation of the MISR on-board calibration panels, that is subsequently applied to panel degradation in the other spectral bands for all cameras (Bruegge et al., 2007).

## 4 Validation of MISR RA *Chl* retrievals against SeaBASS, and comparisons with MODIS

Collocation of the MISR and SeaBASS observations is of course critical to achieving meaningful comparisons. So for each SeaBASS-MISR coincidence, the corresponding location within a MISR orbit is identified as a block (180 blocks per orbit), line (128 along-track lines per block), and sample (512 across-track samples per block) at 1.1 × 1.1 km. We run the RA, as described in section 2.2, over three blocks of data per coincidence, centered on the MISR block that contains the MISR-SeaBASS coincidence. We then interpolate the MODIS-Terra *Chl* data, as well as the associated flags, to the MISR grid via nearest-neighbor interpolation. We flag the following conditions:



- Any MODIS data where the MODIS *Chl* flag data is masked (at level 3) according to http://oceancolor.gsfc.nasa.gov/cms/atbd/ocl2flags.
- Any MISR/MODIS data where the MISR aerosol retrieval acceptance criterion is violated. In this case the criterion, $\chi^2 > 1.0$, is calculated over the all four wavelengths for all glint-free cameras (see section 2.1 above).
- Any MISR/MODIS data where MISR 446nm AOD > 1.0. AOD above this value over ocean tends to occur only in cases of dust, smoke, or pollution plumes, or unmasked clouds. As the surface signal is very small for these cases (especially for the off-nadir cameras), MISR should have little or no sensitivity to *Chl* in these situations.
- Any MISR/MODIS data where the MISR *Chl* $\chi^2 > 1.0$, calculated over the over the blue and green "A" and "B" glint-free cameras.
- Any MISR/MODIS data where in situ *Chl* > 1.5 mg m$^{-3}$.

For comparisons with SeaBASS, we average (in $\log_{10}$ space) up to 5×5 MISR 1.1 km /MODIS 1 km *Chl* retrievals centered on the SeaBASS location, and compare each of the MISR and MODIS-Terra *Chl* to the corresponding SeaBASS value. We also average together the MISR and MODIS results over the same locations. Following conventional practice, $\log_{10}$ of MISR, MODIS, and SeaBASS *Chl* data is taken before any statistics are computed except the mean relative error (MRE).

### 4.1 Validation against SeaBASS

Figure 6 shows three sets of scatterplots for MISR, MODIS-Terra, and the mean of MISR and MODIS, all vs. SeaBASS coincident *Chl*. Points left of the black vertical line in Figure 6 and Table 2 demonstrate MISR sensitivity to retrieving Chlorophyll-a when the in situ value is less than 1.5 mg m$^{-3}$. Statistics for $Chl_{in\ situ} \leq 10$ in Table 2 are shown for completeness. Referring to Table 2, statistics for the 49 SeaBASS coincidences that meet our criteria indicate that the MISR RA performs almost as well as MODIS Terra for these cases. Formally, the average of MISR and MODIS (in $\log_{10}$ space) produce the best overall agreement with SeaBASS: MAE decreases by 29% as compared to MODIS alone, RMSE decreases by 5%, MRE decreases by 17%, and r remains unchanged. However, given the small sample size, it is not possible to draw strong conclusions about whether MISR could in general add value to the MODIS Terra Ocean color product in regions where MODIS-Terra reports *Chl*, despite the likelihood that MISR aerosol retrieval constraints would produce a more accurate atmospheric correction. However, MISR can add value in the glint-contaminated portion of the MODIS-Terra orbit, and in regions of medium-high aerosol loading, where aerosol-type information could improve surface retrieval results (e.g., *Kahn et al.*, 2016).

### 4.2 Comparison against MODIS-Terra

Because the SeaBASS validation dataset contains very few matchups with MISR, in part due to the relatively narrow MISR swath, we compare MISR 1.1 km *Chl* retrievals with collocated MODIS 1 km *Chl* retrievals over much larger regions surrounding the MISR-SeaBASS coincidence locations, using the method described above. We compare to MODIS-Terra



for this regional-context exercise due to the assessments already performed on these data with the much larger number of MODIS-SeaBASS coincidences (e.g., *Franz et al.*, 2012). As such, we compare the MISR RA *Chl* data with all valid pixels for which MODIS *Chl* ≤ 1.5 mg m$^{-3}$.

Figure 7 shows comparisons between the MISR RA and MODIS-retrieved *Chl*, for MODIS *Chl* <10.0 mg m$^{-3}$. The black vertical line indicates 1.5 mg m$^{-3}$. Statistics for the MISR-MODIS *Chl* comparisons, as a function of MODIS-retrieved Chl, are summarized in Table 3. Overall, Figure 7 and Table 3 indicate that the agreement between MISR and MODIS is much better than the agreement between either MISR or MODIS and SeaBASS. The agreement between MISR and MODIS is especially good up until a MISR retrieved *Chl* of 0.5. For MISR *Chl* between 0.5 and 3.0, Figure 7 indicates that a scale

factor could be applied to the MISR data to bring the data into better agreement with MODIS (and likely SeaBASS as well). Comparing MISR vs. MODIS for MODIS *Chl*<1.5: r is 0.05 higher than MODIS vs. SeaBASS, Mean Absolute Error is 36 % lower, RMSE is 32% lower, and MRE is 57% lower. This suggests one or more of the following: (1) MISR-MODIS *Chl* errors co-vary (which is probable to some degree), (2) *Chl* variability is important on these temporal/spatial scales, or (3) we need more in situ data to obtain robust statistics. Regardless, Figures 6 and 7 indicate that there is skill in the MISR *Chl*

retrieval, which could be exploited.

## 5. Conclusions

In Limbacher and Kahn (2014), we detailed extensive modifications to the RA that reduced the 0.024 AOD high bias for AOD$_{558nm}$ < 0.10 to ~0.01 or less. The modifications also improved the results of the RA in general, compared to a set of about 1,100 coincidences with ground-truth observations (lower RMSE, etc.). In Limbacher and Kahn (2015), we

implemented a stray-light correction for the near-nadir cameras based on empirical image analysis with MODIS that further reduced the remaining high bias at low AOD and also improved statistical comparisons to the validation data overall. Here, we performed a radiometric trend analysis over three stable, relatively homogeneous desert sites to identify and quantify radiometric drift in each of the 36 MISR channels. We then applied the radiometric drift corrections to the MISR data in general, further refined the stray-light corrections for the nadir-viewing camera, and applied the stray light corrections to the

other cameras. Finally, we revised the MISR retrieval algorithm to include a chlorophyll-a retrieval, which is implemented so results can be derived from single or multiple MISR cameras.

Justification for the new corrections is as follows: The radiometric trend analysis shows consistent, decreasing BRFs over time for three stable desert sites that can easily be corrected. Errors due to stray-light in the MISR were formally addressed in Limbacher and Kahn (2015), and the adjustments we make in this paper to our ghosting model better represent the stray-

light observed in MISR-MODIS comparisons. These adjustments also allow the corrections to be performed as a series of convolutions, which substantially reduces the ghosting-correction implementation time compared to the approach in Limbacher and Kahn (2015). However, the corrections would run even faster and require fewer approximations if performed



earlier in the MISR data stream, at L1B1, rather than with the L1B2 data available to us, i.e., prior to data rotation, de-convolution, and trimming near the poles.

Validation of the MISR RA-retrieved *Chl,* with all radiometric corrections applied, was performed by comparison with coincident SeaBASS in situ observations. Additionally, comparisons were made against the MODIS-Terra ocean color *Chl*

retrievals because of the relatively small MISR-SeaBASS coincident dataset. Results show that the MISR RA can retrieve *Chl* reliably if the MODIS reported $Chl \leq 1.5$ mg m$^{-3}$, which represents a large fraction of the Earth's ocean area. Compared to SeaBASS, for in situ *Chl* values $\leq 1.5$ mg m$^{-3}$, MISR (MODIS) reports a correlation coefficient of 0.86 (0.91), MAE is 0.10 (0.14), RMSE is 0.25 (0.22), and MRE is 0.52 (0.54), indicating MISR agrees with SeaBASS nearly as well as MODIS Terra when in situ $Chl \leq 1.5$, though for only the 49 available coincidences. For the larger (n=1,499,610) MISR-MODIS

dataset with MODIS-retrieved $Chl \leq 1.5$ mg m$^{-3}$, we find r=0.96, MAE=0.09, RMSE=0.15, and MRE=0.23, indicating that the agreement between MISR and MODIS is substantially better than the agreement between either-instrument and SeaBASS. Differences between these statistics could be explained by one or more of three factors: (1) the number of MISR-SeaBASS coincidences is too small to reach robust conclusions, (2) the temporal/spatial variability of chlorophyll-a is substantial for the 3-hours or the 5x5 box over which the comparisons are made, or (3) the errors in MISR- and MODIS-

Terra-retrieved *Chl* co-vary. Although we find that the MISR RA as implemented here lacks much sensitivity to retrieved *Chl* above 1.5 mgm$^{-3}$, this result was anticipated, due to the lack of spectral bands between 446 and 558 nm (Diner et al, 1998). However, with further work, adjustments to the scattering and absorption in Equation (1) might improve the results in the higher *Chl* regime, particularly if MODIS-Terra reflectances can be integrated into the algorithm.

Obtaining MISR *Chl* retrievals can help fill in the glint-contaminated regions in the single-view MODIS-Terra swath near

the solar equator. In addition, these MISR *Chl* results are derived self-consistently with aerosol amount and type in a physical retrieval, which from the ocean color perspective provides a more robust "atmospheric correction" for the surface retrieval. This work formally opens the door for the use of MISR data in ocean color, complementing the better-constrained and more extensive spectral coverage of MODIS ocean color retrievals. With the improved ocean-surface boundary condition, the MISR multi-angular data should also allow for better-constrained aerosol products, particularly non-sphericity

and single-scattering albedo. A few detailed examples of individual RA joint surface and atmosphere retrievals are given in the Supplemental Material. In the future, it might be possible to ingest collocated MISR and MODIS-Terra reflectances, and use the strengths of each instrument in a complimentary manner.

**Acknowledgments**

We thank Chris Proctor and NASA's Ocean Biology Processing Group for providing the MODIS Terra Ocean Color

products and the SeaBASS group (and cruise PIs) for compiling and providing their in situ ocean color data sets. We thank our colleagues on the Jet Propulsion Laboratory's MISR instrument team and at the NASA Langley Research Center's Atmospheric Sciences Data Center for their roles in producing the MISR Standard data sets, and Brent Holben at NASA




Goddard and the AERONET team for producing and maintaining this critical validation dataset. We also thank Carol Bruegge, Sergey Korkin and Andrew Sayer for *many* helpful discussions, Alexei Lyapustin for providing his MODIS radiometric correction code, as well as Andrew Sayer, James Butler, and Carol Bruegge for comments on an early version of the manuscript. This research is supported in part by NASA's Climate and Radiation Research and Analysis Program under

H. Maring, and NASA's Atmospheric Composition Program under R. Eckman.

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





**Table 1**: Decadal trend values (in percent) aggregated over three stable desert sites for the 36 MISR channels

| Fit | Df | Cf | Bf | Af | An | Aa | Ba | Ca | Da |
|---|---|---|---|---|---|---|---|---|---|
| Blue | -1.03 | -1.22 | -0.85 | -1.14 | -0.22 | -0.44 | -0.68 | -0.37 | -0.20 |
| Green | -1.22 | -1.28 | -1.21 | -1.47 | -1.34 | -1.12 | -1.00 | -0.82 | -0.63 |
| Red | -1.13 | -1.20 | -1.22 | -1.42 | -1.51 | -1.24 | -1.08 | -0.95 | -0.80 |
| NIR | -1.15 | -1.24 | -1.29 | -1.46 | -1.49 | -1.43 | -1.29 | -1.22 | -1.16 |
| | | | | | | | | | |
| **95% CI Fit** | Df | Cf | Bf | Af | An | Aa | Ba | Ca | Da |
| Blue | 0.39 | 0.33 | 0.26 | 0.20 | 0.18 | 0.18 | 0.22 | 0.27 | 0.29 |
| Green | 0.28 | 0.21 | 0.17 | 0.15 | 0.14 | 0.14 | 0.16 | 0.19 | 0.22 |
| Red | 0.17 | 0.13 | 0.11 | 0.09 | 0.09 | 0.10 | 0.11 | 0.13 | 0.17 |
| NIR | 0.17 | 0.12 | 0.10 | 0.09 | 0.08 | 0.09 | 0.10 | 0.13 | 0.21 |
| # | 1186 | 1186 | 1185 | 1158 | 1131 | 1180 | 1168 | 1173 | 1172 |

**The first four rows present the decadal trends for all 4 MISR wavelengths and 9 cameras. The second four rows represent the 95% Confidence Intervals (CI) for the corresponding trends. The final row gives the number of events for each camera.**



**Table 2: Statistics of Chlorophyll-a retrievals as compared to SeaBASS**

| $Chl_{in\ situ}$ <1.5 | r | MAE | RMSE | Fr. Err | # |
|---|---|---|---|---|---|
| MISR RA | 0.86 | 0.10 | 0.25 | 0.52 | 49 |
| MODIS | 0.91 | 0.14 | 0.22 | 0.54 | 49 |
| MISR RA + MODIS | 0.91 | 0.10 | 0.21 | 0.45 | 49 |
| $Chl_{in\ situ}$ <10.0 | r | MAE | RMSE | Fr. Err | # |
| MISR RA | 0.78 | 0.18 | 0.37 | 0.57 | 75 |
| MODIS | 0.88 | 0.16 | 0.26 | 0.52 | 75 |
| MISR RA + MODIS | 0.86 | 0.15 | 0.29 | 0.46 | 75 |

**In this table, r is the Pearson correlation coefficient, MAE is the median absolute error, RMSE is the root mean squared error between the satellite retrieval and in situ data, Fr. Err is the mean absolute fractional error of the retrieval with respect to the** 5 **measurement, and # is the number of validation cases included. The last three rows represent the statistics of an averaged MISR/MODIS retrieval.**





**Table 3: Statistics of MISR vs. MODIS Regional Chlorophyll-a retrievals**

| MISR *Chl* $\chi^2$<1.0, MISR $\chi^2$<1.0 | r | MAE | RMSE | Fr. Err | # |
|---|---|---|---|---|---|
| MODIS *Chl* < 1.5 | 0.96 | 0.09 | 0.15 | 0.23 | 1499610 |
| MODIS *Chl* < 10 | 0.94 | 0.11 | 0.20 | 0.29 | 1829153 |

**In this table, r is the Pearson correlation coefficient, MAE is the median absolute error, RMSE is the root mean squared error between MISR and MODIS-Terra, Fr. Err is the mean absolute fractional error of the MISR RA retrieval with respect to MODIS-Terra, and # is the number of validation cases included.**





**Dark Water Research Algorithm Flow Chart**

For a 17.6 x 17.6 km region, retrieve **TOA model reflectances** ($\varrho_{model}$) and **upward atmospheric transmittances** ($T_{a,up}$) from 8-dimensional LUT. For each coarse grid AOD (12), mixture (74), wavelength $\lambda$(4), and camera(9); *linearly interpolate $\mu_0$(20), WS(5), P(2), $\mu$(16), $\Delta\phi$ (variable)* to the observation viewing/solar geometry, CCMP wind speed, and a pressure of 1013.25 mb.

For each **17.6x17.6 km region**          For each **1.1x1.1 km pixel**

Using LUT, interpolate to observed geometry/wind (for Chl=[0.03,0.1,0.3,1,3,10]): **RfQ=R($\mu$,WS) x f/Q($\mu_0$,$\mu$,$\Delta\phi$,$\lambda$,Chl)**. Calculate the *bidirectional water-leaving radiance $L_w^+$($\mu_0$,$\mu$,$\Delta\phi$,$\lambda$,$\tau$,mix,Chl) = E_d($\mu_0$,$\lambda$,$\tau$,mix) x **RfQ** x b_b($\lambda$,Chl)/a($\lambda$,Chl)*. *Correct $\rho_{model}$ for under-light effects based on $L_w^+$ x $T_{a,up}$ for **all 12 AODs, 774 mixtures, and 6 Chl values. $\rho_{model}$=$\rho_{model}$($\lambda$,cam,$\tau$,mix,Chl)***

Apply *temporal trend*, *ghosting*, and *band-to-band calibration adjustments* + *gas corrections* to the corresponding **MISR TOA observations**.

Using the MISR Retrieval Applicability Mask, mask any camera (at 1.1 km resolution) as cloud contaminated if this mask value exceeds 3 for any band.

For use later in the retrieval: determine the $\chi^2$ **acceptance criteria** ($\chi^2_{min}$* 1.25 + 0.15), absolute measurement uncertainty ([0.005, 0.0025, 0.0015, 0.0015]), and relative measurement uncertainty (3% for all channels). Calculate camera weights based on glitter angle and glint strength (based on NIR method in **2.2**).

**Loop over mixture (774), line (16) and sample (16). Skip if cloud-contaminated. All Interpolation performed is linear or bi-linear. Convert AOD and Chl grid to index based grid such that interpolation is done on unit square.**

**Initial $\tau$ guess**: *Assuming Chl = 0.1*, calculate the $\chi^2$ parameter using the *Red* and *NIR* only for each AOD (12) on the coarse grid. $\tau_o$=$\tau$($\chi^2_{min}$). Calculate 1st and 2nd derivatives (**f', f''**) of $\chi^2$ wrt $\tau$ at $\tau_o$. **$\tau$ guess = $\tau_o$ - f'/f''.**

**Initial Chl guess**: *Interpolating $\rho_{model}$ to $\tau$=$\tau$ guess*, calculate the $\chi^2$ parameter using **all channels** for each Chl (6) on the coarse grid. $Chl_o$=Chl($\chi^2_{min}$). Calculate 1st and 2nd derivatives (**f', f''**) of $\chi^2$ wrt Chl at $Chl_o$. **Chl guess = $Chl_o$ - f'/f''**. Calculate $\chi^2_{new}$=$\chi^2$($\tau$ guess, Chl guess). Set $\chi^2_{old}$=9999.

**$\chi^2$ minimization: Do while $\chi^2_{new}$<0.999*$\chi^2_{old}$ and number of iterations < 5. Grid spacing for derivatives set to 0.01 on unit square.**

Set $\chi^2_{old}$=$\chi^2_{new}$. Compute $\chi^2$ values on a 3x3 grid centered around $\tau_{old}$ and $Chl_{old}$. Calculate **Gradient** ($\nabla$) and **Hessian** (**H**) of $\chi^2$($\tau_{old}$,$Chl_{old}$) wrt $\tau$ and Chl. Calculate next guess of $\tau$ and Chl → [$\tau_{new}$,$Chl_{new}$]$^T$=[$\tau_{old}$,$Chl_{old}$]$^T$-**H$^{-1}$**$\nabla$. Calculate $\chi^2_{new}$=$\chi^2$($\tau_{new}$,$Chl_{new}$)

Save minimized $\chi^2$, retrieved **Chl**, retrieved $\tau$, and $\rho_{model}$, for each mixture, line, and sample.

**Mixture selection and aggregation** (for each 1.1km line and sample):
Weight AOD, AOD Fr. Non-Spherical, absorbing AOD, and aggregate model reflectance by 1/($\chi^2$+0.01) for each of the mixtures passing the minimum acceptance criteria described above. Compute aggregate $\chi^2$, aggregate **Chl $\chi^2$**, and other cloud screening parameters from aggregate model reflectance.

**Figure 1**

**Figure 1. Flow chart describing the MISR RA aerosol/*Chl* retrieval process.**





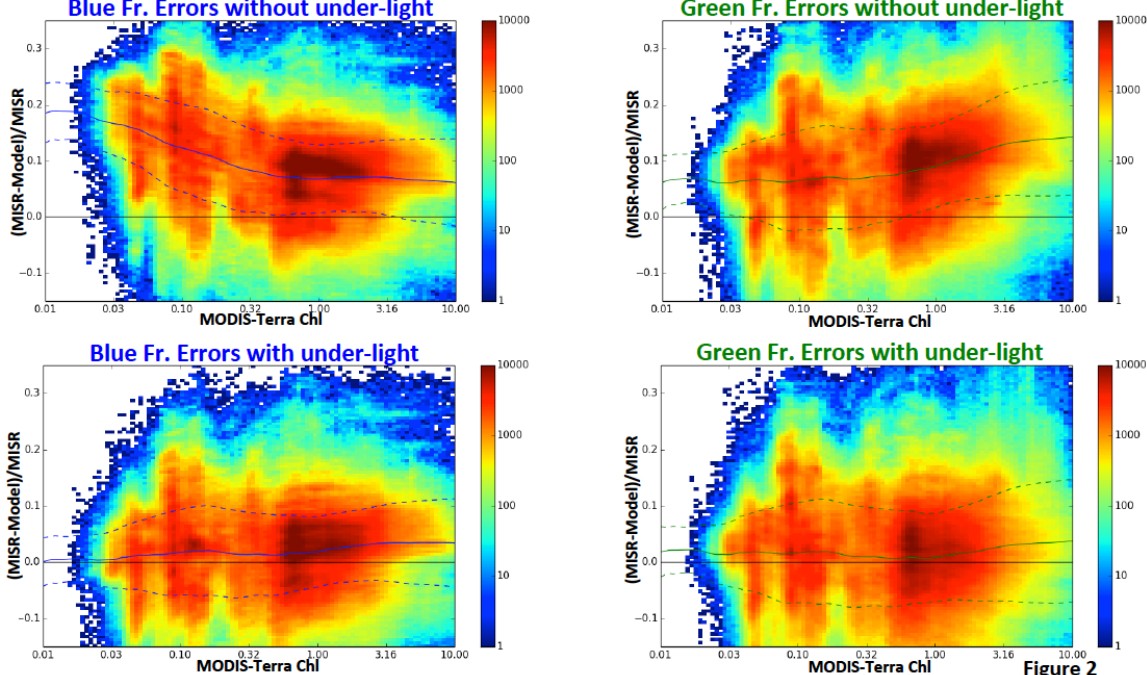

**Figure 2.** The effect of including under-light, assessed by comparing the MISR-observed TOA reflectances with model-simulated values, not including (top panels) and including (bottom panels) under-light calculated with independently retrieved MODIS *Chl* values. These joint histograms show (MISR - Model)/MISR TOA reflectance for the blue (left) and green (right) spectral bands, as a function of MODIS *Chl*. All glint-free cameras are aggregated for this analysis. The solid blue (or green) lines represent the smoothed mean bias, and the dashed lines indicate ± 1 smoothed standard deviation. AOD and mixture were obtained by running the RA with under-light included, based on the MODIS *Chl*, and finding the best-fitting mixture and AOD (using only the NIR band, but up to 9 cameras). Once AOD and mixture were obtained, the TOA reflectances were calculated with the forward model, both with and without under-light. Results show that including under-light dramatically lowers the bias in both the blue and green bands for all *Chl* up to 10 mgm$^{-3}$. As expected, because Chlorophyll-a strongly absorbs in the spectral response range of the MISR blue wavelength, the contribution of under-light to the TOA reflectance decreases with *Chl* in the blue, while it increases with *Chl* in the green due to the enhanced scattering from phytoplankton.





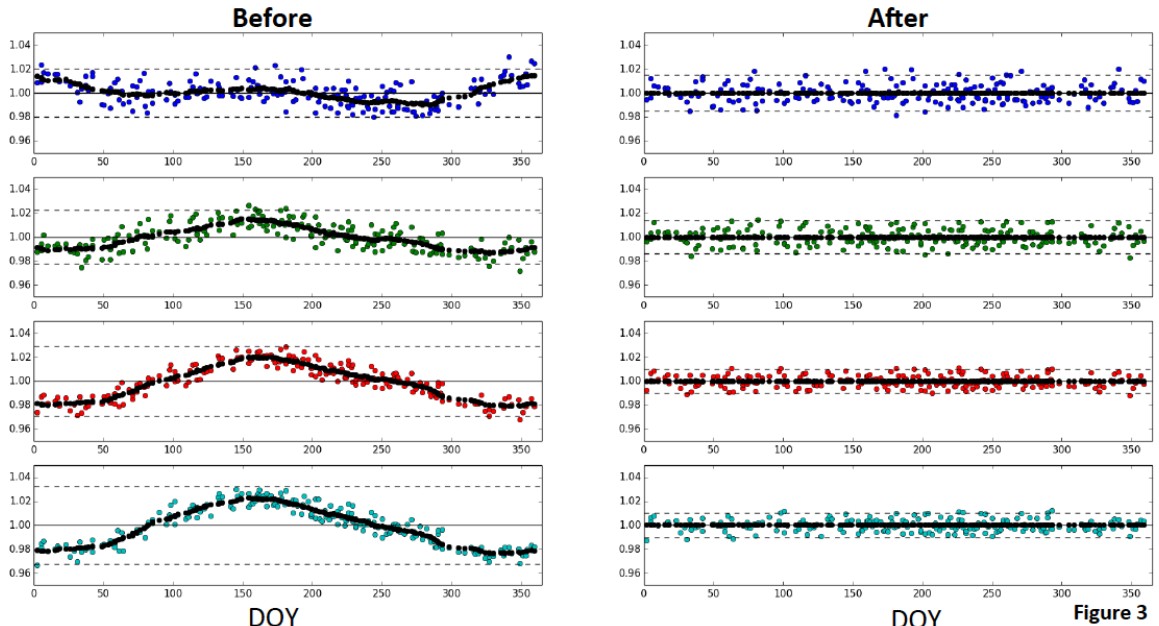

**Figure 3. De-seasonalization example for Libya 4.** Data are normalized such that the mean value of each time-series is unity. Dashed black lines indicate ±2 standard deviations. The plots on the left show the MISR AN (nadir camera) data for the four spectral bands, after Step 2d in Section 3 has been performed. The plots on the right show the same data after Step 3b is complete. These plots present results for only one of two separate paths covering Libya-4, and for only one of nine cameras. Similar analysis was performed for two paths for each of the three stable desert sites.





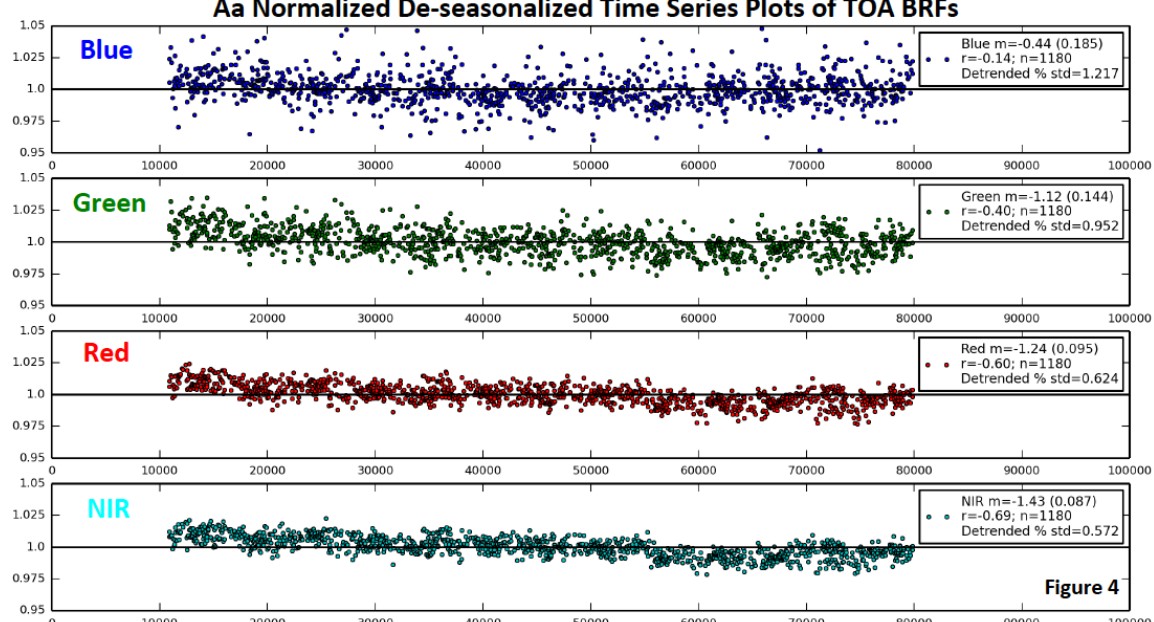

**Figure 4. Normalized, de-seasonalized TOA BRF time series plots, for the four spectral bands of the MISR Aa camera. Data are normalized such that the mean value is unity. These data present all of the data for the three desert sites used (Libya-1, Libya-4, and Egypt-1), excluding outliers, processed through Step 4b of Section 3.**



**Figure 5.** MISR calibration drift per decade (in percent) for all four wavelengths and nine cameras. The data used to generate this plot were aggregated from three pseudo-invariant desert sites (Libya-4, Libya-1, and Egypt-1). The mean decadal trends and the 95% confidence intervals (Student's t-test) are plotted.





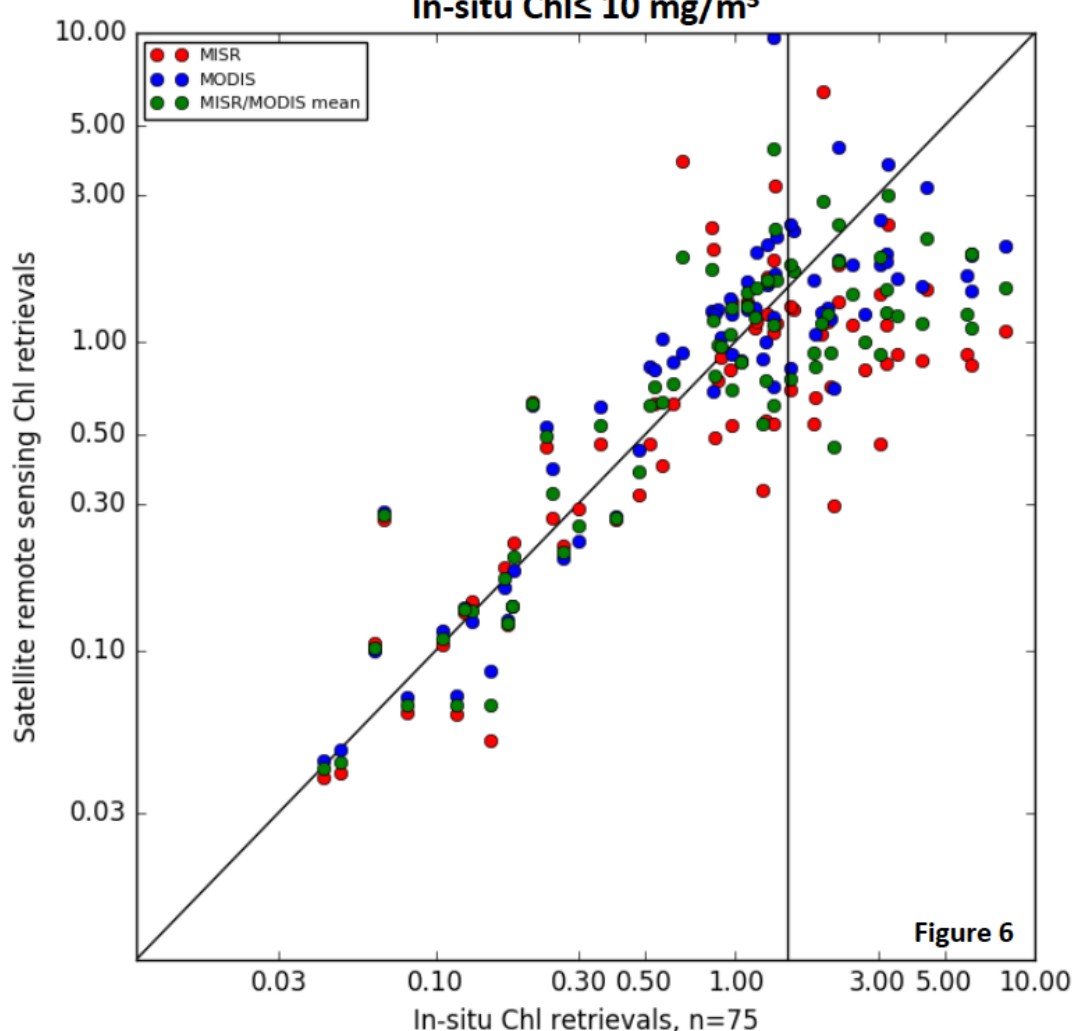

**Figure 6.** MISR (red points), MODIS (blue), and mean MISR/MODIS (green) *Chl* plotted against SeaBASS validation data for *Chl*$_{in\ situ}$ ≤ 10. Results are presented if both MODIS and MISR have at least one valid retrieval in a 5×5 pixel box surrounding the central SeaBASS location. The vertical black line represents *Chl*$_{in\ situ}$ = 1.5.





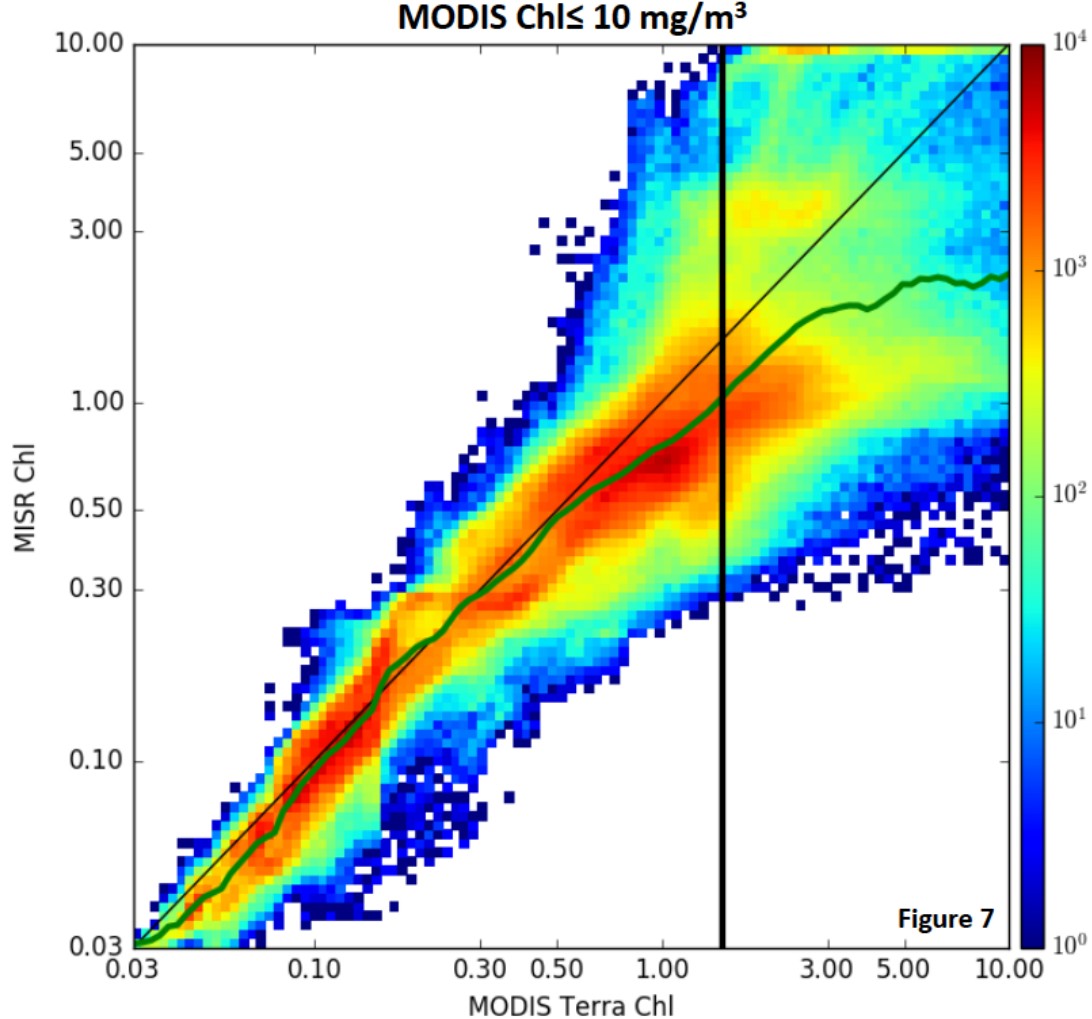

**Figure 7.** MISR-MODIS *Chl* scatter-density plot for $Chl_{MODIS} \leq 10$. The green line represents the mean MISR *Chl* value for each MODIS *Chl* bin, and the vertical black line represents $Chl_{MODIS} = 1.5$. The bin size used for the green line is roughly 0.03 in $\log_{10}$ space.