# Peer review of "Updated MISR Dark-Water Research Aerosol Retrieval Algorithm Part 1: Coupled 1.1 km Ocean-Surface Chlorophyll-a Retrievals with Empirical Calibration Corrections"

_Atmospheric Measurement Techniques, 2016_

## Referee Comment (RC1) · Anonymous Referee #2 · 6 Jan 2017

Overall, I think this is a good paper that merits publication. The authors address the capability of MISR to account for ocean reflectance by parameterizing Chlorophyll-a concentration in the water. Considering the long history of MISR, this was overdue for an investigation.

However, there are some organizational, conceptual and methodological problems in the paper that mask its merits. I think addressing these issues will make for a paper this is more useful to the community (and citable).

[Figure]

One of my main problems is with how the paper is organized, and the relative weight given to describing tweaks to MISR calibration and the incorporation of Chl-a in the Research Algorithm (RA). These are two entirely separate issues, and honestly I think the audience for the latter is far larger than the former (especially considering the literature by the authors and others about calibration in the last few years). This paper feels more like a progress report of all activities performed in a period of time, which isn't the best way to present results to the community. I think this would be a far stronger paper if the calibration changes were either published elsewhere, or in an addendum or the supplementary material. At the very least, a clearer separation in the paper between topics related to calibration and the coupled Chl-a retrieval is needed.

I'm also concerned with the methods used to validate the Chl-a coupled retrieval. Comparisons with SeaBASS are an obvious way of doing things, but of course the number of match-ups available for comparison are limited. What bothers me is the 'validation' against MODIS results. As the authors note, the multi-angle MISR data have access to more information about aerosols, while MODIS has channels better suited for Chl-a retrievals. Differences between coupled algorithm MODIS and MISR results could be due to either instrument – MODIS is not necessarily a standard to which MISR should be held. What is the value, then, of a scatterplot of MODIS and MISR Chl-a results?

Continuing with a discussion coupled Chl-a retrievals, why is there no attempt to determine the uncertainty in such a result based on the components of the RA? SeaBASS data uncertainties are mentioned at one point, but why are those uncertainties are also not incorporated into any validation?

There are a number of statistical tools available to test the hypothesis that dataset A is identical to dataset B. These simple tools go beyond comparisons of correlation coefficients, etc. and actually state the confidence intervals for agreement, taking into account things such as sample size. A problematic example is in the abstract, where it is noted that MODIS has a higher correlation coefficient (0.91) than MISR (0.86). Given the small sample size, these differences are probably statistically insignificant.

Please get an statistics textbook, learn some hypothesis testing techniques, and start applying them. I've also found the following publication to be useful even if it comes from outside our discipline: Altman, D. G. & Bland, J. M. (1983). Measurement in medicine: the analysis of method comparison studies. The statistician, 307–317.

I also wonder why the authors never tested the new RA coupled algorithm on synthetic data. Given the limited SeaBASS dataset, doesn't at least confirming the new algorithm can successfully operate on synthetic data have merit?

Presumably this is part one of a two (or more?) part series? I think how this paper fits in that series needs to be discussed.

Detailed comments

Page 1, line 19-21: I fully do not understand the first sentence of this paragraph – you are running the algorithm to validate the algorithm and somehow also analyze corresponding MODIS data? I think this becomes clearer later on in the paper, but at this point this serves more to confuse than illuminate

Page 1, line 21-23: Does it really make sense to compare a correlation coefficient of 0.91 to 0.86, especially for a small dataset? I think a better way of saying this is that they are statistically identical... which brings me again to the point that you should be using hypothesis testing.

Page 1, line 25: it's not clear at this point the meaning of looking only at Chl<1.5

Page 1-2, final abstract sentence: While I agree that this might be the case, I'm not sure you've demonstrated this in this paper, particularly for the value of joint MISR-MODIS retrievals.

Page 2, line 13: an appropriate continuation of the last sentence of this paragraph would be "assuming that ocean and aerosol signals at TOA don't co-vary."

Page 3, line 27: It would be good to give either a description or a reference for where

the MISR SA wind data come from, and not assume all readers inherently know this.

Page 3, line 27: Why doesn't MISR use NCEP or other modeled products for sea surface pressure? Is that what the SA does, and you're now setting this to 1013.25 because of... mountains near the ocean?

Page 4, lines 2-4: I think separate weighting for aerosol and ocean components of a retrieval would be difficult to implement practically, as the appropriate weights are most likely scene dependent.

Page 5, lines 20-30: I'm really not a fan of arbitrary/empirical weighting algorithms that are presented as fact without any description of the logic behind the choices that were made and their expected significance.

Page 8, line 7: Please use a proper citation for that ATBD or relevant paper, not a website.

Page 9, line 2: "Spectral coefficients of variation" – this is imprecise wording. IS that the CoV for all of one channel for a patch, added up for all channels?

Page 9, lines 16-23: Does this means surface BRDF is assumed to be isotropic? Is that the case?. Also units should be given for the equation on line 17.

Page 10, lines 1-10: Isn't any long term linear trend determined in 3a? Or is this done a year at a time? It is unclear to me from your description.

Page 10, lines 22-24: While I'm sure you're familiar with the Bruegge paper, others may not be. This sentence is quite confusing on its own and needs a more detailed explanation.

Page 11, five starting bullets: all of these are arbitrary choices. It would be nice if you had an explanation for your logic in choosing them.

Page 11, five starting bullets: I'm still not following the logic of why MODIS data should be used to screen MISR results, if that is indeed what is happening (its not clear).

Page 11, line 10: Here you say you flag out Chl > 1.5, but this is presented in the figures? Do the figures show flagged results removed, or not?

Page 11, line 16: what is the value of comparing the mean of MISR and MODIS data to SeaBASS?

Section 4.1: again, I think there are statistical tools you can use that do hypothesis tests that account for sample size. Then you can say in a more quantitative way that the amount of comparison data is "too few" (if it is).

Section 4.1: You mention in several points that collocated MISR data could improve MODIS retrievals of Chl-a. While I agree that this is probably the case, nothing in the analysis you've presented in this paper can demonstrate that.

Page 12, lines 1-3: Again, comparing MODIS and MISR doesn't necessarily indicate the correctness of the MISR algorithm. I could envision some situations (probably with low AOT) where MODIS would work better, and others where MISR would. I think if you had parsed Fig 7. to scenes with high and low AOT, or varying Chl-a, you could start to illuminate these issues.

Page 12, line 14-15: I agree with this

Page 13, line 11: "substantially better" Is this the case, or is this just a symptom of correlation coefficients, etc. calculated with far more cases than the SeaBASS comparison.

Page 13, line 19: Are coupled retrievals performed for MISR scenes where some cameras observe glint, or are they only computed when all cameras observe a scene? If it's the former, do these retrievals work as well since there are fewer angles? Presumably capability would be degraded – how would this affect the ability to fill in MODIS Terra data in glint?

Figure 1: I really had a hard time understanding this flowchart. Please try to make it more legible. Minimize text where you can – less is more.

Figure 2: It would be nice to remind readers the actual wavelength of the blue and green channels

Figures 4-5: Nobody knows what "Camera Aa", etc. means outside the MISR group. Please just state the viewing angle instead.

Supplementary material

Figure S1: Can you see the colored value of the AERONET sites? I can't.

Only here to you allude to a "part II" of this study. I can only assume that portion will have more details about the coupled RA. Which makes me think that there should really just be two papers, one about calibration, one about coupled retrievals.

All supplementary figures: I'm confused why there are consistently fewer MODIS retrievals in some of the scenes, can this be discussed?

---

## Referee Comment (RC2) · A. Lyapustin (Referee) · 15 Feb 2017

[referee-annotated manuscript omitted]

---

## Author Response (AR1)

The authors would like to thank Alexei Lyapustin for his feedback. As a result, we have made some changes to the structure of the manuscript.

This paper describes an updated MISR research algorithm for aerosol and chlorophyll retrieval over Case 1 waters. The improvements include a standard explicit model of underlight as a function of ChI, and, importantly, improvements in calibration, including new de-trending analysis particularly important for climate research applications. This is a solid work that needs to be published with minor revision.

General comment: While generally the paper is written reasonably well, an improvement in structure/logic would be very helpful. Currently, the text unfolds almost unstructured as a story: the details of algorithm are mixed with calibration, past and new work, and at some point it becomes rather confusing as to what new is specifically done here.

It could help if you could structure these things upfront in Introduction.

We have restructured the paper such that we believe it now flows more clearly.

Page 2, lines 31-32: Ocean reflectance in the blue can be higher than 5-20%; as far as I know, Gordon referred to Red band with respect to relative contribution.

Blue albedo in case I waters appears to peak at about 20% according to Figure 2 from our paper. The 5-20% is from our work here, and might be specific to the MISR channels.

Page 3, lines 21-22: Be more specific: which model is used - is it isotropic Cox-Munk, or Nakajima-Tanaka, or something else?

We have made it clear in the text that the model is isotropic Cox-Munk.

Page 3, line 28: How is the MISR Standard product surface pressure aliased from nearby mountains to over ocean?

As a single surface pressure value is used for a 17.6 km region, and we do retrievals at 1.1 km resolution, it is possible for errors to manifest themselves as we approach the coast (if there are mountains nearby).

Page 3, line 29: To use all 4 bands, you need to accurately limit to Case-1 waters. How is it done?

The  $\chi^2$  parameter (calculated over all 4 wavelengths) and our  $\chi^2_{Chl}$  parameter should be effective at limiting results to only case I. Because the *Chl* parameterization is based on a case I framework, our model-measurement fits

should be very poor in case II conditions; this is true for the cases we've tried. This is now mentioned in the text.

Page 8, line 28: Lyapustin et al. technique does full atmospheric correction of MODIS TOA data, contrary to this approach, as described. Thus, the core assumption that the AOD is assumed temporally stable, should be explicitly mentioned.

We agree, and have added that the AOD is assumed to be temporally stable.

Page 12, line 7: To give reader a good reference frame for MISR data, can you provide the same numbers for the SeaWIFS - Terra comparison (these are widely available).

We could provide the numbers in a table (as a Terra-SeaWIFS comparison on the SeaBASS website takes only a few seconds to compute), but they would be for a completely different sample subset, and so might not be comparable to the data presented. However, we have commented on comparisons between MODIS-Terra, SeaWIFS, and SeaBASS in the text, including mentioning that SeaWIFS agrees better with MODIS-Terra than MISR does.

**Reviewer 2**

The authors would like to thank reviewer 2 for their feedback. As a result, we have made some modifications to the manuscript.

One of my main problems is with how the paper is organized, and the relative weight given to describing tweaks to MISR calibration and the incorporation of Chl-a in the Research Algorithm (RA). These are two entirely separate issues, and honestly I think the audience for the latter is far larger than the former (especially considering the literature by the authors and others about calibration in the last few years). This paper feels more like a progress report of all activities performed in a period of time, which isn't the best way to present results to the community. I think this would be a far stronger paper if the calibration changes were either published elsewhere, or in an addendum or the supplementary material. At the very least, a clearer separation in the paper between topics related to calibration and the coupled Chl-a retrieval is needed.

Because of the spectrally varying nature of MISR's calibration (e.g., the green band has degraded much faster than the blue), it makes sense to address the calibration in this paper. As the reviewer is well aware, there is a high degree of correlation between ratios of blue-to-green water leaving radiance and retrieved chlorophyll-a. Therefore, any calibration (or stray-light) artifacts that affect this ratio need to be taken into account before the analysis is performed. We have included this point in the main text, and moved the calibration section into an appendix. I'm also concerned with the methods used to validate the Chl-a coupled retrieval. Comparisons with SeaBASS are an obvious way of doing things, but of course the number of match-ups available for comparison are limited. What bothers me is the 'validation' against MODIS results. As the authors note, the multi-angle MISR data have access to more information about aerosols, while MODIS has channels better suited for Chl-a retrievals. Differences between coupled algorithm MODIS and MISR results could be due to either instrument – MODIS is not necessarily a standard to which MISR should be held. What is the value, then, of a scatterplot of MODIS and MISR Chl-a results?

We agree that this is more a comparison against MODIS than a formal validation, as we make clear in the text. However, the MODIS product is a standard for the ocean color community, and its properties have been studied by many groups. For example, the SeaBASS website gives error statistics for MODIS-Terra, which we also note in the paper. As such, comparing MISR to MODIS-Terra is useful, in part because they share the same platform, in part because MODIS ChI is a better-known and substantially validated quantity, and in part because there are vastly more MISR-MODIS coincidences than MISR-SeaBASS coincidences. Further, we note that before presenting the MISR-MODIS comparison, we present the MISR-SeaBASS comparison, for which there are only 49 points.

Continuing with a discussion coupled Chl-a retrievals, why is there no attempt to determine the uncertainty in such a result based on the components of the RA? SeaBASS data uncertainties are mentioned at one point, but why are those uncertainties are also not incorporated into any validation?

The RA is a coupled atmospheric and surface algorithm compiled into a selfconsistent aerosol/ChI retrieval algorithm with appropriate spectral and angular weighting. As there is limited information on the separate uncertainty of each component, we assess the uncertainty for the coupled system. We have added error bars (for SeaBASS) to the scatter-plots.

There are a number of statistical tools available to test the hypothesis that dataset A is identical to dataset B. These simple tools go beyond comparisons of correlation coefficients, etc. and actually state the confidence intervals for agreement, taking into account things such as sample size. A problematic example is in the abstract, where it is noted that MODIS has a higher correlation coefficient (0.91) than MISR (0.86). Given the small sample size, these differences are probably statistically insignificant.

We agree that the MISR-SeaBASS comparison is unlikely to be statistically significant, and we perform a two-sample Kolmogorov-Smirnov test to indicate that the MISR-MODIS results cannot be distinguished from each other, with better than 95% certainty. We made the MISR-SeaBASS comparison for completeness, as the SeaBASS data are considered ground truth, but we also made the MISR-MODIS comparison, as mentioned earlier, in part to address the SeaBASS-MISR sampling limitations. We have modified the abstract, validation, and conclusions to improve the paper and address these points.

Please get an statistics textbook, learn some hypothesis testing techniques, and start applying them. I've also found the following publication to be useful even if it comes from outside our discipline: Altman, D. G. & Bland, J. M. (1983). Measurement in medicine: the analysis of method comparison studies. The statistician, 307–317.

We are of course familiar with the standard hypothesis tests, such as the Student's T-test, and even with some more advanced tests that do not assume a Gaussian form for the underlying distributions. We have applied a two sample Kolmogorov-Smirnov test to assess the distinguishability of datasets in the paper. However, the statistics are still provided in Tables 1 and 2.

I also wonder why the authors never tested the new RA coupled algorithm on synthetic data. Given the limited SeaBASS dataset, doesn't at least confirming the new algorithm can successfully operate on synthetic data have merit? Presumably this is part one of a two (or more?) part series? I think how this paper fits in that series needs to be discussed.

As the MISR-SeaBASS dataset is so limited, a key part of our motivation for the MISR-MODIS comparison. As discussed above, this allows us to compare MISR to MODIS-Terra's already validated Chl. The focus of our research program in general is aerosols, so our first objective here is to improve the lower boundary condition over ocean. If a MISR-derived Chl product proves to be of value to the community despite the reduced coverage and sub-optimal spectral bands compared to MODIS, we would consider further Chl validation in a future study.

**Detailed comments**

Page 1, line 19-21: I fully do not understand the first sentence of this paragraph – you are running the algorithm to validate the algorithm and somehow also analyze corresponding MODIS data? I think this becomes clearer later on in the paper, but at this point this serves more to confuse than illuminate.

We agree that this is a little confusing. It has been rewritten.

Page 1, line 21-23: Does it really make sense to compare a correlation coefficient of 0.91 to 0.86, especially for a small dataset? I think a better way of saying this is that they are statistically identical which brings me again to the point that you should be using hypothesis testing.

The MISR-SeaBASS comparison is performed solely because SeaBASS is a standard used for ocean color validation. The MISR-MODIS comparison is included specifically to obtain adequate statistics; we now make it clear in the text that the MISR and MODIS data cannot be distinguished using a two-sample Kolmogorov-Smirnov test.

**Page 1, line 25: it's not clear at this point the meaning of looking only at Chl<1.5**

We have clarified our reasoning behind this by indicating that this is the region we expect meaningful results from our ChI model. It is also the regime where the empirical, oligotrophic water ChI algorithm used by much of the ocean color community has its best performance [e.g., Hu et al., JGR 2012].

Page 1-2, final abstract sentence: While I agree that this might be the case, I'm not sure you've demonstrated this in this paper, particularly for the value of joint MISR-MODIS retrievals.

We believe our statement is worded appropriately. We are just indicating that one of MISR's strengths is its ability to diagnose aerosol type (due to having multiple camera angles), and that a future MISR-MODIS retrieval algorithm could leverage this strength appropriately. Similarly, we note that MODIS provides superior spectral information content for this application.

Page 2, line 13: an appropriate continuation of the last sentence of this paragraph would be "assuming that ocean and aerosol signals at TOA don't co-vary."

Right, we added that.

Page 3, line 27: It would be good to give either a description or a reference for where the MISR SA wind data come from, and not assume all readers inherently know this.

You are correct, and we have added a reference to a personal communication with Mike Garay, who currently maintains the SA. Basically, the wind comes from monthly Quickscat (prior to ~2011) and SSMI (2011 onwards) wind speed data.

Page 3, line 27: Why doesn't MISR use NCEP or other modeled products for sea surface pressure? Is that what the SA does, and you're now setting this to 1013.25 because of mountains near the ocean?

Many of the decision about ancillary inputs to the SA were made more than a decade ago, when computer resources, storage, throughput, etc., were quite limited compared to today. To run the SA operationally on the entire MISR datastream at the time, the SA uses one value of monthly-modeled pressure for each 17.6 km x 17.6 km region. This can result in over-ocean errors in the

RA retrievals where the elevation changes dramatically near the coast. Given the small variations in surface sea-level pressure relative to the mean over cloud-free scenes where the over-water retrievals are performed, using 1013.25 mb as the surface pressure is reasonable.

Page 4, lines 2-4: I think separate weighting for aerosol and ocean components of a retrieval would be difficult to implement practically, as the appropriate weights are most likely scene dependent.

In general, the weighting for the ocean surface component could be  $\mu$ , whereas the aerosol component could be  $\mu^{-1}$ . This makes use of the greater surface contribution to the TOA reflectance, relative to the atmospheric contribution, in the near-nadir cameras, and conversely, the greater atmospheric contribution to the steeper-angle signals. However, as we have not implemented this aerosol/ocean weighting algorithm yet (as was noted in the text), we have removed our reference to it.

Page 5, lines 20-30: I'm really not a fan of arbitrary/empirical weighting algorithms that are presented as fact without any description of the logic behind the choices that were made and their expected significance.

Although we agree with your assessment on the appearance of arbitrary numbers related to our glint screening, the glint mask used previously in the MISR RA, also the SA, the MODIS and other satellite aerosol-retrieval algorithms, are also arbitrary, are static, and are therefore yet more restrictive. The new weighting is adaptive; it represents a substantial improvement to the original logic and also to the results. We have added the following to the paper: "The minimum and maximum reflectances were taken via forward modeling, and 25° was set as the minimum because, as glitter angle decreases, a small error in wind speed could substantially impact the retrieval. This is a conservative choice; as we improve our ability to determine if a camera is glint-contaminated in the future, we will likely lower this minimum glitter-angle threshold."

Page 8, line 7: Please use a proper citation for that ATBD or relevant paper, not a website.

The citation for the algorithm itself was referred to on line 3, but it may not have been obvious what OBPG refers to; we have clarified this in the references. The website gives a very good description of the algorithm, so we are also leaving the link in the text.

Page 9, line 2: "Spectral coefficients of variation" – this is imprecise wording. IS that the CoV for all of one channel for a patch, added up for all channels?

We explain this on the following line. 'The spectral coefficients of variation (standard deviation divided by the mean) were calculated for rolling  $50 \times 50$  pixel patches, separately in each spectral band of the nadir camera...'.

Page 9, lines 16-23: Does this means surface BRDF is assumed to be isotropic? Is that the case?. Also units should be given for the equation on line 17.

This equation is only a scaling of the TOA radiances to produce TOA reflectances, and it retains the angular dependence of the scene. The units of BRF are sr-1.

Page 10, lines 1-10: Isn't any long term linear trend determined in 3a? Or is this done a year at a time? It is unclear to me from your description.

We subtract out the trend before deseasonalization, which removes the seasonal noise from the signal, and add it back afterward. This allows for a result with much lower uncertainty. We have added this point explicitly to the text.

Page 10, lines 22-24: While I'm sure you're familiar with the Bruegge paper, others may not be. This sentence is quite confusing on its own and needs a more detailed explanation.

This has been clarified with the following:

"The MISR calibration procedure assumes that the panel degrades in a spectrally invariant way. This is likely a poor assumption that produces a spectrally varying TOA reflectance drift with time."

Page 11, five starting bullets: all of these are arbitrary choices. It would be nice if you had an explanation for your logic in choosing them.

We have added some explanation, but disagree that the choices are arbitrary; they are motivated by the need to use only the highest-quality data for this comparison.

Page 11, five starting bullets: I'm still not following the logic of why MODIS data should be used to screen MISR results, if that is indeed what is happening (its not clear).

MODIS-Terra provides a validated and widely used ocean color product coincident with MISR observations. Here we compare collocated data points for an apples-to-apples comparison. The text now also makes clear that "we mask out any MISR/MODIS data where the MODIS ChI flag data is masked...".

Page 11, line 10: Here you say you flag out Chl > 1.5, but this is presented in the figures? Do the figures show flagged results removed, or not?

We do present all data with ChI < 10 mg m-3, this has been corrected in the text. We tend to focus on the ChI <1.5 mg m-3, as we expect much more sensitivity to retrieved ChI in this regime. This is borne out in the plots.

Page 11, line 16: what is the value of comparing the mean of MISR and MODIS data to SeaBASS?

We have removed the MISR-MODIS mean results from the paper.

Section 4.1: again, I think there are statistical tools you can use that do hypothesis tests that account for sample size. Then you can say in a more quantitative way that the amount of comparison data is "too few" (if it is).

We have commented on this above, and make clear the sample-size limitations.

Section 4.1: You mention in several points that collocated MISR data could improve MODIS retrievals of ChI-a. While I agree that this is probably the case, nothing in the analysis you've presented in this paper can demonstrate that.

Providing additional information content to either MISR or MODIS should improve the retrieval. Furthermore, on physical grounds, we expect that the improved "atmospheric correction" would also improve the ocean surface retrieval, particularly near coasts and along aerosol transport pathways over ocean [e.g., Kahn et al., 2016].

Page 12, lines 1-3: Again, comparing MODIS and MISR doesn't necessarily indicate the correctness of the MISR algorithm. I could envision some situations (probably with low AOT) where MODIS would work better, and others where MISR would. I think if you had parsed Fig 7. to scenes with high and low AOT, or varying ChI-a, you could start to illuminate these issues.

Given a sufficiently large sample size and an uncertainty envelope (obtained from SeaBASS) for MODIS-Terra, one could use MODIS-Terra for a more formal validation. As discussed above, the MISR-MODIS comparison is actually more compelling than the MISR-SeaBASS comparison, because of the very small MISR-SeaBASS sample size, and the validation that has been performed on the MODIS ocean color product. We think stratifying MISR/MODIS ChI results based on AOD and aerosol type is beyond the scope of this paper, but should be considered for future work after validation of the MISR RA AOD and aerosol type is performed.

Page 13, line 11: "substantially better" Is this the case, or is this just a symptom of

correlation coefficients, etc. calculated with far more cases than the SeaBASS comparison.

We have added statistics for the three collocated datasets (MISR, MODIS-Terra, and SeaBASS) to Table 2. We agree it is not possible to definitively say that the statistics are substantially better, and so we have changed the wording accordingly. However, after comparing MODIS-Terra, SeaWIFS, and SeaBASS ChI data from the SeaBASS website, it looks very much like the satellite-remote sensing retrievals are co-varying and that the satellite-remote sensing results agree better with each other than with SeaBASS.

Page 13, line 19: Are coupled retrievals performed for MISR scenes where some cameras observe glint, or are they only computed when all cameras observe a scene? If it's the former, do these retrievals work as well since there are fewer angles? Presumably capability would be degraded – how would this affect the ability to fill in MODIS Terra data in glint?

Yes, coupled retrievals are performed in regions where the nadir (An) camera is in glint. Although we don't have the in-situ comparison to prove this, it is very likely that the uncertainty in the ChI retrieval is tied to the number of cameras used in the retrieval once the glint-contaminated cameras are removed (as well as the viewing geometry of those cameras). The ChI retrieval will probably not be impacted at all by the loss of a "D" camera (70.5° viewing zenith), but the loss of all 3 near-nadir "A" cameras will almost certainly result in higher uncertainty. This is now noted in the text.

Figure 1: I really had a hard time understanding this flowchart. Please try to make it more legible. Minimize text where you can – less is more.

We have learned from reviews of earlier papers that putting the algorithm detail in a flow-chart allows for better understanding of the algorithm with some readers. The structure of the flow chart, and the bolding of some text, is aimed at helping the reader sort out the information. As most readers view our papers digitally these days, an interested reader can zoom in to more easily see the detail, as needed.

Figure 2: It would be nice to remind readers the actual wavelength of the blue and green channels

The numbers are mentioned earlier in the paper, but we have added them here as well.

Figures 4-5: Nobody knows what "Camera Aa", etc. means outside the MISR group. Please just state the viewing angle instead. We have indicated what the A, B, C, and D camera terminology means in the caption.

**Supplementary material**

**Figure S1: Can you see the colored value of the AERONET sites? I can't.**

Yes, we can see it. Because the AERONET values are mapped to the same color scale as MISR, a good match will be difficult to see. Indeed, for the RA, AOD, ANG, Fr. Non-Sph, and SSA all match quite well to AERONET. We have however adjusted the plots so they are easier to interpret.

Only here to you allude to a "part II" of this study. I can only assume that portion will have more details about the coupled RA. Which makes me think that there should really just be two papers, one about calibration, one about coupled retrievals.

We think the calibration is necessary for this work, as discussed above, but we have put it in an appendix.

All supplementary figures: I'm confused why there are consistently fewer MODIS retrievals in some of the scenes, can this be discussed?

MISR runs down the center of the MODIS swath. As the center of the MODIS swath tends to be in sun-glint anywhere near the solar equator, a substantial portion of MISR's swath will have no quality controlled MODIS-Terra ChI retrievals. Because MISR uses up to 9 cameras over ocean, there are always at least a few that will not be contaminated by sun-glint. We have clarified this in the supplemental material.

Updated MISR Dark-Water Research Aerosol Retrieval Algorithm

**Part 1: Empirical Calibration Corrections and Coupled 1.1 km Ocean-Surface Chlorophyll-a Retrievals with Empirical Calibration Corrections**

James A. Limbacher1,2 and Ralph A. Kahn1

1Earth Science Division, NASA Goddard Space Flight Center

, Greenbelt MD, 20771, USA

2Science Systems and Applications Inc., Lanham-MD, 20706. USA

For: Atmospheric Measurement Technologies

Submitted: October 2016

**Style Definition:** Normal: Font: 10 pt, Justified, Line spacing: 1.5 lines

Style Definition: Header: Font: 10 pt, Justified, Line spacing: 1.5 lines, Tab stops: 8 cm, Centered + 16 cm, Right + Not at 7.62 cm + 15.24 cm

**Style Definition:** Hyperlink: Font color: Blue

**Style Definition:** List Paragraph: Font: 10 pt, Justified, Line spacing: 1.5 lines

**Style Definition:** Balloon Text: Font: (Default) Tahoma, 8 pt, Justified

Style Definition: Footer: Font: 10 pt, Justified, Tab stops: 7.96 cm, Centered + 15.92 cm, Right + Not at 7.62 cm + 15.24 cm

**Correspondence to: Ralph A. Kahn (ralph.a.kahn@nasa.gov)**

**Abstract**

- As aerosol amount and type are key factors in the "atmospheric correction" required for remote-sensing chlorophyll-aconcentration (Chl) retrievals, the Multi-Angle Imaging SpectroRadiometer (MISR) can contribute to ocean color analysis despite a lack of spectral channels optimized for this application. Conversely, an improved ocean-surface constraint should also improve MISR aerosol-type products, especially spectral single-scattering albedo retrievals. We introduce a coupled, self-consistent retrieval of Chl together with aerosol over dark water. There are time-varying MISR radiometric calibration errors that significantly affect key spectral reflectance ratios used in the retrievals. Therefore, we also develop and apply new calibration corrections to the MISR top-of-atmosphere (TOA) reflectance data, and introduce a self consistent retrieval of Chl The calibration corrections include: a modified stray-light model based on with acrosol over dark water together comparisoncomparisons, with coincident MODerate resolution Imaging Spectroradiometer (MODIS) Terra data, observations and trend analysis usingof the MISR TOA bidirectional reflectance factors (BRFs) over three pseudo-invariant desert sites-The trend analysis shows that MISR radiometric sensitivity decreased by up to 2 percent for MISR spectral bands between January 2002 and December 2014.

After applying calibration corrections, we We run the MISR Research Retrieval Algorithm (RA) to validatewith the corrected MISR RAreflectances to generate MISR-retrieved *Chl*, and analyze bothcompare the MISR and corresponding MODIS Terra-*Chl* values compared to a set of 49 collocatedcoincident SeaBASS *in-\_situ* observations, constrained to-, Where *Chl*n-situ < 1.5 mgm gm  $^{-3}$ . 
[revised manuscript text omitted]

$$\begin{aligned} \underline{L}_{w}^{+}\left(\underline{m}_{0},\underline{m},\underline{\mathsf{D}f},/,\underline{WS},\underline{t},\underline{mix},\underline{Chl}\right) &= \underline{E}_{d}\left(\underline{m}_{0},/,\underline{t},\underline{mix}\right)^{*} \Re\left(\underline{m},\underline{WS}\right)^{*} \left(\begin{array}{c} \underline{b}_{b}\left(/,Chl\right)\\ a\left(/,Chl\right)\end{array}\right)^{*} \frac{f}{Q}\left(\underline{m}_{0},\underline{m},\underline{\mathsf{D}f},/,Chl\right)\\ \underline{L}_{w}^{+}\left(\underline{m}_{0},\underline{m},\underline{\mathsf{D}f},/,WS,\underline{t},\underline{mix},Chl\right) &= \underline{E}_{d}\left(\underline{m}_{0},/,\underline{t},\underline{mix}\right)^{*} \Re\left(\underline{m},WS\right)^{*} \left(\begin{array}{c} \underline{b}_{b}\left(/,Chl\right)\\ a\left(/,Chl\right)\end{array}\right)^{*} \frac{f}{Q}\left(\underline{m}_{0},\underline{m},\underline{\mathsf{D}f},/,Chl\right) \underbrace{\qquad} (1) \end{aligned}$$

[revised manuscript text omitted]

 9                                                                                                                                                                                                                                                                                                                                                                                                                                                                                                                                                                                                                                                                                                                                                                                                                                                                                                                                                                                                                                                                                                                                                                                                                                                                                                                                                                                                                                                                                                                                                                                                                                                                                                                                                                                 |   |                       |   |
| eameras at this step.               |                                                                                                                                                                                                                                                                                                                                                                                                                                                                                                                                                                                                                                                                                                                                                                                                                                                                                                                                                                                                                                                                                                                                                                                                                                                                                                                                                                                                                                                                                                                                                                                                                                                                                                                                                                                                                                                                                   |   |                       |   |
| 2) Remove outliers for a            | ach noth/site and spectral band                                                                                                                                                                                                                                                                                                                                                                                                                                                                                                                                                                                                                                                                                                                                                                                                                                                                                                                                                                                                                                                                                                                                                                                                                                                                                                                                                                                                                                                                                                                                                                                                                                                                                                                                                                                                                                                   |   |                       |   |
| -,                                  | avad madian PDFs by acquisition data fit a line to the values and subtrast the linear                                                                                                                                                                                                                                                                                                                                                                                                                                                                                                                                                                                                                                                                                                                                                                                                                                                                                                                                                                                                                                                                                                                                                                                                                                                                                                                                                                                                                                                                                                                                                                                                                                                                                                                                                                                             |   |                       |   |
| trend from th                       | e data.                                                                                                                                                                                                                                                                                                                                                                                                                                                                                                                                                                                                                                                                                                                                                                                                                                                                                                                                                                                                                                                                                                                                                                                                                                                                                                                                                                                                                                                                                                                                                                                                                                                                                                                                                                                                                                                                           |   |                       |   |
| <del>b. Aggregate th</del>          | e data by day of year (DOY) and smooth the sorted, de trended BRFs using a 21-                                                                                                                                                                                                                                                                                                                                                                                                                                                                                                                                                                                                                                                                                                                                                                                                                                                                                                                                                                                                                                                                                                                                                                                                                                                                                                                                                                                                                                                                                                                                                                                                                                                                                                                                                                                                    |   |                       |   |
| <del>point (i.e., ±1</del>          | 0 data point) rolling average. (The data are sufficiently dense that replacing each data                                                                                                                                                                                                                                                                                                                                                                                                                                                                                                                                                                                                                                                                                                                                                                                                                                                                                                                                                                                                                                                                                                                                                                                                                                                                                                                                                                                                                                                                                                                                                                                                                                                                                                                                                                                          |   |                       |   |
| <del>point with the</del>           | e mean of 21 points does not create significant artifacts in the time-series.)                                                                                                                                                                                                                                                                                                                                                                                                                                                                                                                                                                                                                                                                                                                                                                                                                                                                                                                                                                                                                                                                                                                                                                                                                                                                                                                                                                                                                                                                                                                                                                                                                                                                                                                                                                                                    |   | armattad. Dight. 0 am | _ |
| <del>C.a. Identity BRF</del> | 5 diat fuil outside 26 from time series.                                                                                                                                                                                                                                                                                                                                                                                                                                                                                                                                                                                                                                                                                                                                                                                                                                                                                                                                                                                                                                                                                                                                                                                                                                                                                                                                                                                                                                                                                                                                                                                                                                                                                                                                                                                                                                          |   | ormatted: Right: 0 cm | _ |
| d. b. Remove the             | identified outliers from the original data.                                                                                                                                                                                                                                                                                                                                                                                                                                                                                                                                                                                                                                                                                                                                                                                                                                                                                                                                                                                                                                                                                                                                                                                                                                                                                                                                                                                                                                                                                                                                                                                                                                                                                                                                                                                                                                       |   |                       |   |
| This step removed 3-149             | t of data outliers from each time series.                                                                                                                                                                                                                                                                                                                                                                                                                                                                                                                                                                                                                                                                                                                                                                                                                                                                                                                                                                                                                                                                                                                                                                                                                                                                                                                                                                                                                                                                                                                                                                                                                                                                                                                                                                                                                                         |   |                       |   |
| 3) <del>De-seasonalize the da</del> | ta for each site and spectral band                                                                                                                                                                                                                                                                                                                                                                                                                                                                                                                                                                                                                                                                                                                                                                                                                                                                                                                                                                                                                                                                                                                                                                                                                                                                                                                                                                                                                                                                                                                                                                                                                                                                                                                                                                                                                                                |   |                       |   |
| a. Fit a line to the                | ne original, time-ordered BRFs, with outliers removed, and linearly de-trend the data.                                                                                                                                                                                                                                                                                                                                                                                                                                                                                                                                                                                                                                                                                                                                                                                                                                                                                                                                                                                                                                                                                                                                                                                                                                                                                                                                                                                                                                                                                                                                                                                                                                                                                                                                                                                            |   |                       |   |
| b. Re aggregate
rolling average  | the data by DOY and smooth the BRFs again using a 21 point (±10 data point)
<del>ze.</del>                                                                                                                                                                                                                                                                                                                                                                                                                                                                                                                                                                                                                                                                                                                                                                                                                                                                                                                                                                                                                                                                                                                                                                                                                                                                                                                                                                                                                                                                                                                                                                                                                                                                                                                                                                                     |   |                       |   |
| e.a. Rearrange the                  | • data by time and add back the linear trend from Step 3a.                                                                                                                                                                                                                                                                                                                                                                                                                                                                                                                                                                                                                                                                                                                                                                                                                                                                                                                                                                                                                                                                                                                                                                                                                                                                                                                                                                                                                                                                                                                                                                                                                                                                                                                                                                                                                        | F | ormatted: Right: 0 cm |   |
| Step 3 is illustrated in Fi         | <del>gure-3 for the Libya 4 site.</del>                                                                                                                                                                                                                                                                                                                                                                                                                                                                                                                                                                                                                                                                                                                                                                                                                                                                                                                                                                                                                                                                                                                                                                                                                                                                                                                                                                                                                                                                                                                                                                                                                                                                                                                                                                                                                                           |   |                       |   |
| 4) Normalize the data               | •                                                                                                                                                                                                                                                                                                                                                                                                                                                                                                                                                                                                                                                                                                                                                                                                                                                                                                                                                                                                                                                                                                                                                                                                                                                                                                                                                                                                                                                                                                                                                                                                                                                                                                                                                                                                                                                                                 | F | ormatted: Right: 0 cm |   |
| a. Normalize the                    | e data so the time series mean for each spectral band at each site is 1.0, allowing data                                                                                                                                                                                                                                                                                                                                                                                                                                                                                                                                                                                                                                                                                                                                                                                                                                                                                                                                                                                                                                                                                                                                                                                                                                                                                                                                                                                                                                                                                                                                                                                                                                                                                                                                                                                          |   |                       |   |
| from multiple                       | e sites and paths to be compared.                                                                                                                                                                                                                                                                                                                                                                                                                                                                                                                                                                                                                                                                                                                                                                                                                                                                                                                                                                                                                                                                                                                                                                                                                                                                                                                                                                                                                                                                                                                                                                                                                                                                                                                                                                                                                                                 |   |                       |   |
| The result is 216                   | normalized time series, one for each MISR camera and band, for each of two paths at three4                                                                                                                                                                                                                                                                                                                                                                                                                                                                                                                                                                                                                                                                                                                                                                                                                                                                                                                                                                                                                                                                                                                                                                                                                                                                                                                                                                                                                                                                                                                                                                                                                                                                                                                                                                                        | F | ormatted: Right: 0 cm |   |
| sites.                              |                                                                                                                                                                                                                                                                                                                                                                                                                                                                                                                                                                                                                                                                                                                                                                                                                                                                                                                                                                                                                                                                                                                                                                                                                                                                                                                                                                                                                                                                                                                                                                                                                                                                                                                                                                                                                                                                                   |   |                       |   |
| b. These time seri                  | ies are then aggregated across all paths to produce 36 time series, one for each MISR channel                                                                                                                                                                                                                                                                                                                                                                                                                                                                                                                                                                                                                                                                                                                                                                                                                                                                                                                                                                                                                                                                                                                                                                                                                                                                                                                                                                                                                                                                                                                                                                                                                                                                                                                                                                                     |   |                       |   |
| (Figure 4)                          |                                                                                                                                                                                                                                                                                                                                                                                                                                                                                                                                                                                                                                                                                                                                                                                                                                                                                                                                                                                                                                                                                                                                                                                                                                                                                                                                                                                                                                                                                                                                                                                                                                                                                                                                                                                                                                                                                   |   |                       |   |
| (8)                                 |                                                                                                                                                                                                                                                                                                                                                                                                                                                                                                                                                                                                                                                                                                                                                                                                                                                                                                                                                                                                                                                                                                                                                                                                                                                                                                                                                                                                                                                                                                                                                                                                                                                                                                                                                                                                                                                                                   |   |                       |   |
| The linear percent shapes per       | denote and its 05 th percent confidence interval are then calculated for each shannel                                                                                                                                                                                                                                                                                                                                                                                                                                                                                                                                                                                                                                                                                                                                                                                                                                                                                                                                                                                                                                                                                                                                                                                                                                                                                                                                                                                                                                                                                                                                                                                                                                                                                                                                                                                  |   |                       |   |
| The initial percent change per      | The second contraction of the second contract of the second |   |                       |   |

[revised manuscript text omitted]